# Ultra-wideband optical coherence elastography from acoustic to ultrasonic frequencies

Xu Feng [1,3], Guo-Yang Li[1,3] & Seok-Hyun Yun [1,2] ✉

Visualizing viscoelastic waves in materials and tissues through noninvasive imaging is valuable for analyzing their mechanical properties and detecting internal anomalies. However, traditional elastography techniques have been limited by a maximum wave frequency below 1-10 kHz, which hampers temporal and spatial resolution. Here, we introduce an optical coherence elastography technique that overcomes the limitation by extending the frequency range to MHz. Our system can measure the stiffness of hard materials including bones and extract viscoelastic shear moduli for polymers and hydrogels in conventionally inaccessible ranges between 100 Hz and 1 MHz. The dispersion of Rayleigh surface waves across the ultrawide band allowed us to profile depth-dependent shear modulus in cartilages ex vivo and human skin in vivo with sub-mm anatomical resolution. This technique holds immense potential as a noninvasive measurement tool for material sciences, tissue engineering, and medical diagnostics.

Measurement of the mechanical properties of materials is routinely performed in many places in sciences, engineering, and industries, as well as hospitals[1–4]. Widely used tools include strain–stress testing[5] and dynamic mechanical analysis[6] for measuring bulk properties, and atomic force microscopy (AFM)[7,8] and microrheology[9,10] for local measurements. In clinical medicine, elastography has been adopted for disease diagnosis[11–14]. Elastography allows to noninvasively measure the elasticity of tissues in normal and abnormal states using medical imaging modalities, such as ultrasound and magnetic resonance imaging (MRI). In all these tools, samples under test are deformed by some force, their responses are measured, and the mechanical properties are calculated from the data[15,16]. The timescale or frequency range of the measurement spans from quasi static (as slow as $10^{-5}$ Hz) to acoustic (as fast as $10^3$ Hz) ranges. Higher speeds up to a few tens of kHz have been used in AFM[17] and in our recent work on dynamic optical coherence elastography (OCE)[18]. In the latter, elastic waves of short impulse or continuous-wave monotones are generated, their propagation within a tissue is visualized, and from the data the wave velocities are determined and related to the elastic moduli of the tissue[15,19].

While the material analysis thus far has been focused on the quasi-static to acoustic ranges, we hypothesized that a higher frequency range beyond 10 kHz can offer a window of opportunity that was previously underappreciated especially for elastography. First, the higher frequency data can reveal the viscoelastic characteristics of materials in the shorter time scale. Dynamic mechanical analysis (DMA) is widely used to characterize viscoelasticity, but it has a limited frequency range (1–100 Hz). The time-temperature superposition technique can mimic the high-frequency responses of simple, homogeneous materials[20] but is not applicable to composite materials and living matters[9]. Ultrasound non-destructive testing[21,22] is an established technique using ultrasound waves, typically at 500 kHz–20 MHz, to measure the time of flight across a material or structure with well-defined boundaries to determine bulk elastic properties, but this technique is not directly applicable to elastography and especially for complex materials such as tissues. Second, since the spatial resolution of elastography is approximately given by the wavelength of elastic waves[23], the higher frequency can lead to the higher resolution. Third, wave velocity dispersion over an extended frequency range is useful to

[1]Harvard Medical School and Wellman Center for Photomedicine, Massachusetts General Hospital, 50 Blossom St. BAR-8, Boston, MA 02114, USA. [2]Harvard-MIT Health Sciences and Technology, Cambridge, MA 02139, USA. [3]These authors contributed equally: Xu Feng, Guo-Yang Li.
✉e-mail: syun@hms.harvard.edu

extract more detailed mechanical information such as the depth-dependent variation of elasticity and internal stress. The depth information is not accessible using laser Doppler vibrometry or AFM-based rheology that only probes the surface[24–26].

Despite all these anticipated benefits mentioned above, it is technically challenging to extend the upper frequency limit. Because the wave energy is proportional to $A^2f^2$ ($A$, amplitude; $f$, frequency) and the viscosity tends to grow with $f$, the energy dissipation or damping of a wave increases with $A^2f^3$. So, with increasing frequency, the wave amplitude must be reduced by $~f^{3/2}$ to avoid excessive sample heating or damage. Detecting the reduced amplitude requires improved sensitivity, for example, by a factor of a thousand when comparing 100 kHz–1 kHz. To analyze waves with submicron amplitudes, a nanometer-scale sensitivity is required to an instrument. OCE is an emerging elastography technique[11,27] built on optical coherence tomography (OCT)[28]. OCE has been developed with a variety of different system architectures and applied to various tissue types, such as cornea[18,29], sclera[30], breast[31], brain[32], and skin[33]. As OCE uses optical interferometry, it offers superior detection sensitivity in the order of nanometers, compared to micrometers for ultrasound and millimeters for MRI. The high sensitivity of OCE make it a good candidate for elastography beyond the acoustic range.

Here, we demonstrate an OCE system capable of visualizing elastic waves from the acoustic to ultrasonic frequencies up to a few MHz. We developed a novel aliasing technique, demodulation algorithm, and jitter-correction method to handle such high frequencies far beyond the typical axial line-scan (A-line) rates of OCT. After verifying the system with homogeneous materials, we apply it to dynamic shear analysis of soft materials, and stiffness mapping of complex tissues in a knee joint ex vivo and human skin in vivo with layer-resolving resolution. Our results demonstrate the benefits and broad applicability of ultra-wideband OCE.

## Results
### Signal demodulation of high-frequency vibration in swept−source OCE

Figure 1a depicts a basic interferometer used in swept source OCT, where the optical wavenumber is tuned repeatedly with a period of $T$: $k = k_0 + k_1[t]$, where $[t] = mod(t,T) − T/2$, $k_0 = 2\pi/\lambda$ ($\lambda$, center wavelength), and $k_1$ is a tuning rate[34]. Imagine a mirror-like sample that vibrates with an amplitude $\delta$, angular modulation frequency $\omega_m(= 2\pi f_m)$, and reflection coefficient $r$ at a mean depth of $z_0$. As the depth is modulated by $\delta \cos(\omega_m t)$, we obtain an interferometric signal $I(t) = r \quad \cos(2k_0 z_0 + 2k_1 z_0 t + 2k_0 \delta \cos(\omega_m t) + 2k_1 t\delta \cos(\omega_m t))$, where

$2k_1 z_0 T$ and $\omega_m T$ are assumed to be multiples of $2\pi$ for convenience. Figure 1b illustrates $I(t)$ for $\delta \ll \lambda$. For the typical case of $k_1 T \ll k_0$ we get $I(t) \approx r \cos(2k_0 z_0 + \omega_0 t + 2k_0 \delta \cos(\omega_m t))$, where $\omega_0 = 2k_1 z_0$ is the carrier frequency. In the frequency domain, it consists of an amplitude $rJ_0(2k_0\delta) \approx r$ at $\omega_0$ and two first harmonic sidelobes at $\omega_0 \pm \omega_m$ with an amplitude of $rJ_1(2k_0\delta) \approx rk_0\delta$. This is depicted in Fig. 1c. In OCT, the frequency-domain analysis is performed by discrete Fourier transform $F(\omega) = FFT\{I(t)\}$ for each A-line data. The frequency resolution is about equal to the half of the A-line rate $f_A = 1/T$. OCE has been operated with relatively low modulation frequencies less than the A-line rate: that is, $f_m < 0.5f_A$. This ensures that the first harmonic sidelobes overlap with the main peak. The carrier and modulated components interfere with each other in the pixel at $\omega_0$ with a phase $\phi \approx 2k_0 z_0 + 2k_0 \delta \cos(\omega_m t)$. From the amplitude of the phase variation, $\delta$ is readily measured.

It is possible to extend OCE to ultrasonic frequencies simply by employing ultrafast OCT systems with MHz A-line rates[35–37] while satisfying $f_m < 0.5f_A$. Then, the same data processing method as described above is applicable. However, since most OCT systems, including commercial products, use A-line rates less than 100 kHz, it should be worthwhile to develop a general method for OCE that works even when $f_m > f_A/2$. Below we describe such a method, which takes advantage of signal aliasing.

### General demodulation algorithm for high frequency OCE

When $f_m > f_A/2$, the modulation sidelobes separate from the carrier (Fig. 1d). In this case, one cannot directly apply the conventional method. We rewrite $I(t) \xrightarrow{k_0\delta \ll 1} Re\{re^{i2k_0 z_0}[e^{i\omega_0 t} + k_0\delta e^{i(\omega_0 - \omega_m)t} + k_0\delta e^{i(\omega_0 + \omega_m)t}]\}$. Now that the modulation component is separated from the carrier, the vibration amplitude $k_0\delta$ manifest itself in the amplitude of the Fourier component at $\omega_0 \pm \omega_m$. In this case, $\delta$ can be measured from the amplitude, not the phase. However, this is only possible for the mirror-like sample when there is no other signal (background) present at $\omega_0 \pm \omega_m$.

Consider a tissue-like sample that has continuous scattering points along its depth with $r(z)$. The problem of retrieving $\delta$ from the scatterer at $z_0$ becomes more complicated. We used simulated data to demonstrate this problem (Fig. 2). As shown in Fig. 2a, the Fourier component $F(\omega_0 − \omega_m)$ contains three signals: First, the vibration signal (negative sideband) originated from the scatterer at $\omega_0$ (Scatterer 1). Second, the time-independent reflection signal from a different scatterer at $z_0 − z_m$ (Scatterer 2), where $z_m = \omega_m/(2k_1)$; Depending on whether $r(z_0 − z_m) > r(z_0)k\delta$ or not, the phasor circle may or may not exclude the origin, and $\delta$ is embedded differently in the phase and amplitude of $F(\omega_0 − \omega_m)$). Third, the vibration signal (positive

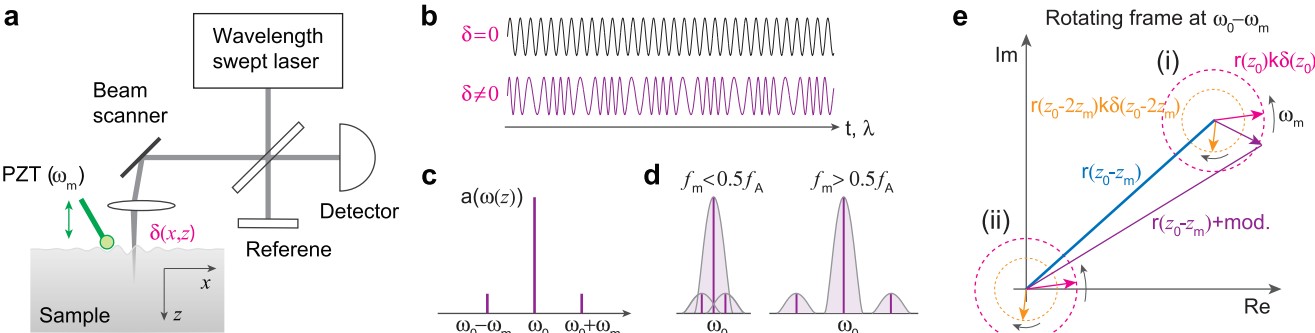

**Fig. 1 | Principle of high-frequency OCE. a** Schematic of a swept-source interferometric setup. A piezoelectric actuator excites elastic waves with angular frequency $\omega_m$ and amplitude profile $\delta(x,z)$. **b** Example of time-domain detector signals without and with vibration of a mirror. The primary carrier frequency is given by the depth location ($z$) of the reflector and the wavelength sweep speed. **c** Frequency-domain transform of the trace in (**b**). **d** Overlap of the carrier and modulation components for low and high frequencies compared to the A-line rate ($f_A$). $f_m$:

modulation frequency, $\omega_0$: carrier frequency. **e** Graphical relationship between $\delta(z)$ and the reflection coefficient $r(z)$ in the Fourier-domain at $\omega_0 − \omega_m$. Depending on whether $r(z_0 − z_m)$ is greater or smaller than the magnitude of rotating phasor $r(z_0)$ $k_0 \delta(z_0)$, the pixel phase undergoes an oscillatory or diverging pattern. For demodulation, the rotating phase circles (i) is brought to the origin (ii) by subtracting the offset $r(z_0 − z_m)$, and $\delta(z_0)$ is determined from the magnitude of the counter-clockwise rotating phasor.

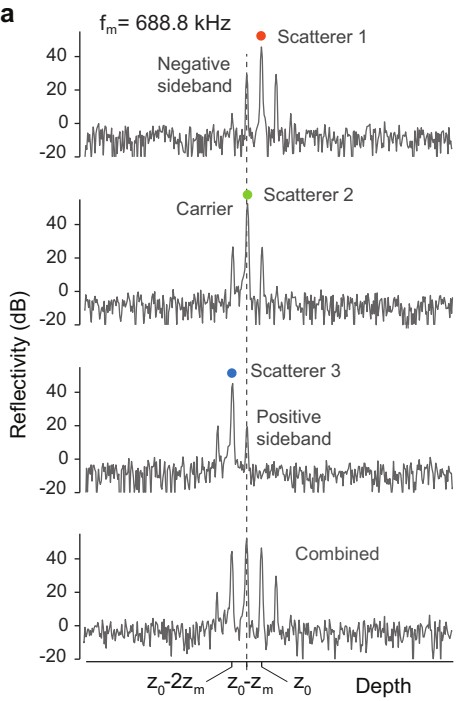

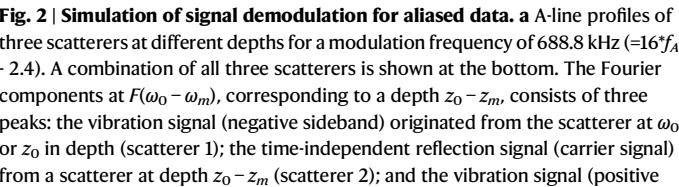

**Fig. 2 | Simulation of signal demodulation for aliased data. a** A-line profiles of three scatterers at different depths for a modulation frequency of 688.8 kHz ($=16{*}f_A$ - 2.4). A combination of all three scatterers is shown at the bottom. The Fourier components at $F(\omega_0 - \omega_m)$, corresponding to a depth $z_0 - z_m$, consists of three peaks: the vibration signal (negative sideband) originated from the scatterer at $\omega_0$ or $z_0$ in depth (scatterer 1); the time-independent reflection signal (carrier signal) from a scatterer at depth $z_0 - z_m$ (scatterer 2); and the vibration signal (positive sideband) originated from the scatterer located at $z_0 - 2z_m$ (scatterer 3). **b** Steps of signal demodulation. (i) Original signal $F(\omega_0 - \omega_m)$. (ii) Static DC background signal $\langle F(\omega_0 - \omega_m)\rangle$. (iii) Subtracting the original signal from the DC background. (iv) Normalizing the signal by the carrier DC background $\langle F(\omega_0)\rangle$. (v) Fourier transform (FFT) to identify $\delta(z_0)$ and $\delta(z_0 - 2z_m)$. Both the real part (red line) and the imaginary part (blue line) of the amplitude plots are shown. $f_A$: A-line rate, $\omega_0$: center wavenumber, $\omega_m$: angular frequency of the vibration signal, $\delta$: vibration amplitude.

sideband) originated from the scatterer located at $z_0 - 2z_m$ (Scatterer 3). The vibration amplitudes of Scatterers 1 and 3 can be retrieved from $F(\omega_0 - \omega_m)$ using a demodulation algorithm.

Figure 2b illustrates the method that involves subtracting the static DC background $\langle F(\omega_0 - \omega_m)\rangle$ from $F(\omega_0 - \omega_m)$, where $\langle\ \rangle$ denotes time average. This brings the phasor circles to the origin as depicted in Fig. 1e (ii). Second, we normalize the signal from the carrier DC $r(z_0) = \langle F(\omega_0)\rangle$. Finally, we identify $\delta(z_0)$ and $\delta(z_0 - 2z_m)$ from a Fourier transform of $F - \langle F\rangle$. The rigorous mathematical description of the demodulation algorithm is provided in Methods. The situation of $f_m > f_A/2$ is equivalent to signal *aliasing* that occurs when the modulation frequency is greater than the sampling rate. In fact, our algorithm works for any $f_m$ values regardless of whether it is aliased or not. Thus, this is a general algorithm for OCE.

## Experimental demonstration of anti-aliasing demodulation

We used a swept-source OCT system previously built using a polygon-scanner wavelength-swept laser providing a center wavelength of 1307 nm, 3-dB bandwidth of 80 nm, axial resolution of 16 μm, A-line rate $f_A$ of 43.2 kHz, and maximum average optical power of 12 mW on samples (Supplementary Fig. S1). We used phase-sensitive interferometric detection[38,39] to obtain sub-wavelength vibration amplitudes. The detector had a bandwidth of 100 MHz, and its output was digitized at 100 MS/s with 14-bit resolution. An input/output (I/O) board was used to generate waveforms for a galvanometer beam scanner. The I/O board or a function generator was used to generate stimulus waveforms applied to the piezoelectric transducer (PZT). During data acquisition, the stimulus waveforms were simultaneously recorded using the I/O board, which was later used for time jitter correction.

To test our demodulation scheme, we used a PZT actuator block and apply a sinusoidal waveform synchronously with a master clock in

the OCT system (Supplementary Fig. S2). To generate mirror-like reflection, a flat glass plate was attached to the PZT. The laser power on the sample was attenuated to generate a signal-to-noise (SNR) of 40 dB at the air-glass interface.

Figure 3a shows a typical A-line profile in the Fourier domain when $f_m$ was 679.6 kHz. The signal appeared at the 107th pixel. The interval between pixels is 21.5 μm in depth or 52.7 kHz. Two first harmonic sidelobes emerged at about 12 pixels away from the main peak. The side lobes were lower by 20 dB in signal power (10 dB in amplitude) than the main lobe. Since the ratio of the sidelobe to the carrier is equal to $k_0\delta$, we obtain $\delta = 20.4$ nm. To verify its accuracy, we characterized the same PZT using a laser Doppler vibrometer. The vibration amplitudes measured by OCE were consistent with those measured by the Doppler vibrometer (see Supplementary Fig. S3). Furthermore, we measured the sideband position over a broad range of excitation frequencies. Our result showed that the sideband position increased linearly with the vibration frequency, with the expected slope of 0.019 pixel/kHz (Fig. 3b).

## Noises and time jitter correction for ultrasonic frequencies

Various noises ultimately limit the system's sensitivity to vibration and the accuracy of wave velocity measurement. The fundamental limit comes from optical SNR, $X$, in A-lines. The length of the modulation-induced phasor is $r(z_0)k_0\,\delta(z_0)$. The minimum detectable amplitude $\delta_{min}$ is then given by $r(z_0)k_0\,\delta_{min}(z_0) = r_{min}(z_0 - z_m)$, where $r_{min}(z_0 - z_m) = r_{min}(z_0) = r(z_0)X(z_0)^{-0.5}$ is the minimum detectable reflection coefficient by definition[40]. So, $k_0\,\delta_{min} = X^{-0.5}$, same for non-aliased and aliased regimes. We note that although $\delta(z_0)$ is obtained from sidebands at $z_0 \pm z_m$, its error is governed by the SNR of the main peak at $z_0$. In OCE, $\delta$ is typically extracted from a set of M-mode data consisting of $N$ A-lines. Then, the sensitivity is improved by $\sqrt{N}$, and $k_0$

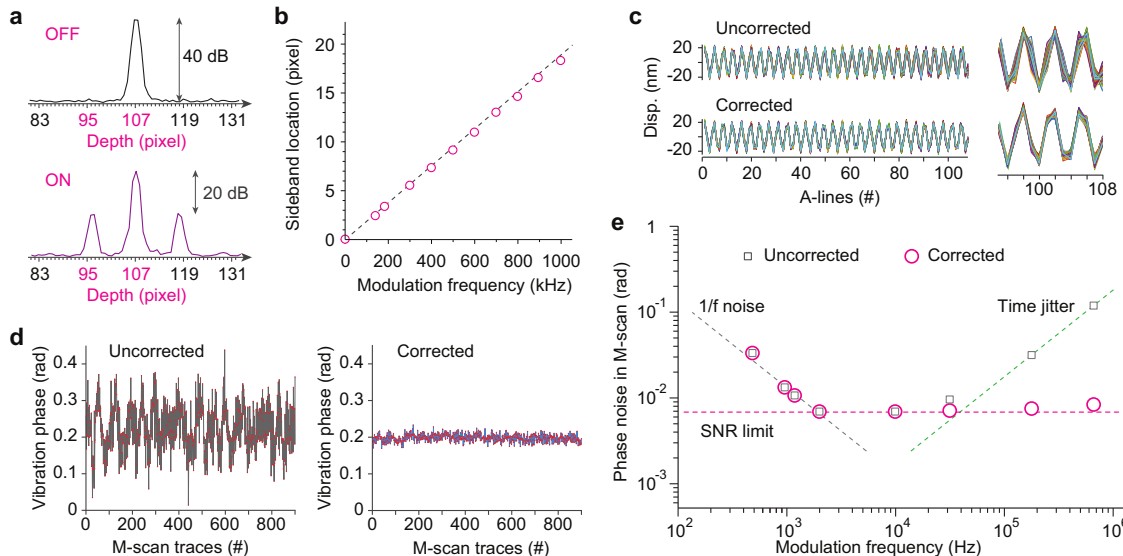

**Fig. 3 | Implementation of aliasing in OCE. a** Signal from the surface of a stationary PZT in the frequency (depth) domain without modulation (top) and with modulation at 679.6 kHz (bottom). **b** Measured position of modulation peak at different modulation frequency and the theoretical slope. **c** 900 M-scan traces at 679.6 kHz before and after time jitter correction. A single M-scan trace consists of 108 A-lines. **d** Phase data of M-scan traces showing noises before and after jitter correction calculated over 900 M-scan traces. **e** Phase noise performance measured with optical SNR of 40 dB and $N = 108$. The SNR-limited phase noise corresponds to $-94$ dB/Hz. The 1/f noise is also seen. Note that this graph is for 40 dB SNR. For lower SNR, the optical phase noise level would increase. For example, for 20 dB optical SNR, the optical noise level is moved up to 0.07 rad, crossing the 1/f noise line at 200 Hz.

$\delta_{min} = X^{-0.5} N^{-0.5}$. For the above experiment, where $N = 108$ and $X = 10^4$, we find $\delta_{min} = 195$ pm. Another fundamental noise source is 1/f electrical noise. SNR at low frequencies may be limited by the 1/f noise rather than the reflectivity-limited phase noise. Mechanical vibrations of optical components add noises. In our system, the mechanical jitters of beam scanners and rotating polygon filters were suppressed to practically acceptable levels.

Additionally, noises can come from the time jitter of the swept-source laser (i.e., small variations of the A-line rate). This time error accumulates through successive sweeps which generates phase noise through electronics boards and signal generators (Fig. 3c). This noise increases in proportion to the modulation frequency and can become dominant at the ultrasonic range. To measure and correct the time jitter in real time, we used the phase of the stimulus waveform, which was measured in situ during data acquisition and contain the same time jitter, to correct the phase of the modulated signal in each M-scan (see the detailed flowchart in Supplementary Fig. S4). This phase correction was effective (Fig. 3d). The total phase noise after correction coincides with the SNR limited noise (Fig. 3e). The stimulus waveform was also used to calculate the laser time jitter. We found the time-jitter over sweep period $T$ follows a Gaussian distribution with a standard deviation $\sigma_T$ of 4.9 ns in our system (Supplementary Fig. S5).

Next, we derive the theoretical time jitter induced phase error $\sigma_L$. Let $\tau_i$ denote the period jitter in the $i$-th sweep, where $\langle \tau_i \rangle = 0$ and $\langle \tau_i - \langle \tau_i \rangle \rangle = \sigma_T$. The timing of the $m$-th A-line is given by $t_m = mT + \sum_{i=1}^{m} \tau_i$. The output from the $m$-th A-line is $F_m = k_0 \delta e^{i\omega_m t_m}$. $\delta$ is determined from the inverse discrete Fourier transform of $F_m$ at $\omega_m$. The result is $k_0 \delta(1 + i\omega_m \sum_{m=1}^{N} \sum_{i=1}^{m} t_i / N)$. The standard variation of the second term is $\sigma_L \approx \omega_m \sigma_T \sqrt{(N+1)(2N+1)/6N^2}$ (Supplementary Note 1). For $N \gg 1$, we get $\sigma_L \approx \sqrt{N/3} \omega_m \sigma_T$. The total phase noise $\Delta\phi$ is a square sum of the SNR- and jitter-induced noises: $\Delta\phi = \sqrt{1/NX^2 + 0.58N(\omega_m \sigma_T)^2}$.

Finally, the phase noise causes uncertainty in determining the wave velocity, $v$, from the phase variation along the propagation distance. The error $\Delta v$ in velocity is given by $\Delta v / v = \Delta\phi / 2\pi(\lambda/L)$, where $L$ is the effective measurement length. In soft materials, elastic waves are attenuated, and $L$ is typically a couple of wavelengths. Therefore, $\Delta\phi = 0.01$ yields a high accuracy of $\Delta v / v < 1\%$.

## Ultrasonic OCE of hard materials

To verify our system at ultra-high frequencies, we measured various hard materials. Mechanical waves were excited on the surface by using a custom-made contact probe with the piezoelectric actuator (see Methods). For a 5 mm-thick acrylate plastic block, we obtained a flat dispersion curve over 1–2 MHz (Fig. 4a). Since the wavelength (<1.4 mm) is smaller than the thickness, the excited wave is the Rayleigh surface wave with a velocity $c_R = (0.862 + 1.14v)/(1 + v)c_s$, where $c_s$ is the pure (bulk) shear wave speed and $v$ denotes the Poisson's ratio. The shear elastic modulus $\mu$ is given by $\mu = \rho c_s^2$, where $\rho$ is the density. From the measured $c_R = 1285 \pm 16$ m/s and known $\rho = 1.2$ g/cm$^3$ and $v = 0.37$, we obtained $\mu = 2.26 \pm 0.06$ GPa. This value at MHz is slightly higher than the literature value of 1.7 GPa measured by quasi-static mechanical tools[41]. The difference is presumably due to the viscoelasticity of the material. Next, we measured a 1 mm-thick polystyrene plate. Figure 4b shows a representative $k$-domain plot measured at 787.6 kHz. The dispersion relations for the A0 and S0 waves were fitted using the Lamb wave model[42]. The result was $\mu = 1.64 \pm 0.05$ GPa, in comparison to quasi-static shear modulus of ~1.3 GPa reported in literature[43]. We also performed OCE on glass and metal (Supplementary Fig. S6). We obtained $\mu = 24.0 \pm 0.61$ GPa for borosilicate glass and $45.7 \pm 1.79$ GPa for copper (Fig. 4c). These values were consistent with their quasi-static shear moduli[44,45], indicating their near-pure elasticity. We also performed OCE on the cortical surface of a bovine tibia bone specimen ($\rho = 1.6$ g/cm$^3$ and $v = 0.3$) and found $\mu$ to vary from $2.36 \pm 0.04$ GPa at 485 kHz to $3.60 \pm 0.12$ GPa at 2 MHz (Fig. 4d). This frequency dependence is in part due to the viscoelasticity of the bone but also contributed by its depth-dependent variation of modulus. Later we will show how the wave velocity dispersion can be used to measure depth-dependent elasticity.

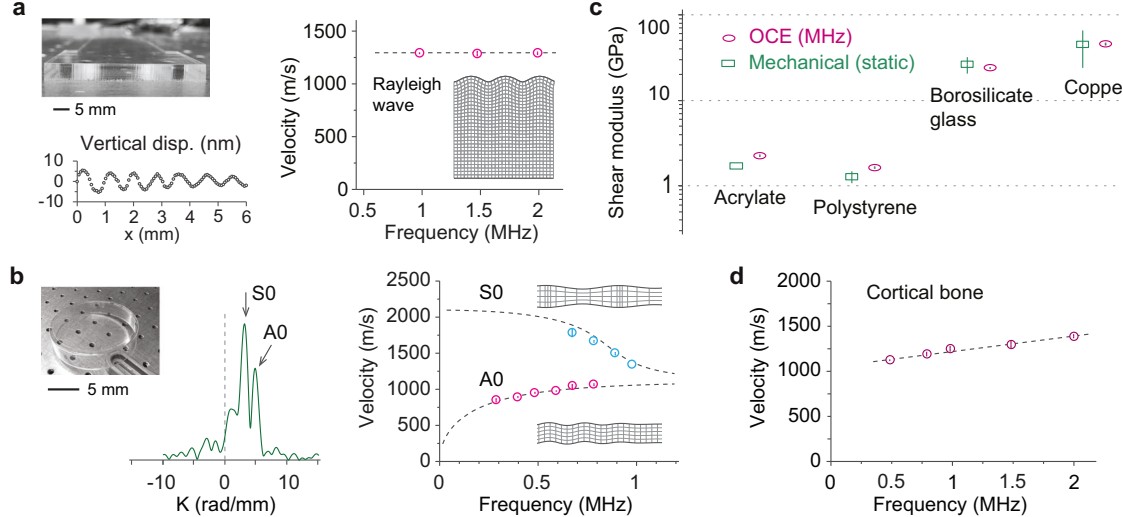

**Fig. 4 | Ultrasonic OCE of hard materials. a** Measured Rayleigh wave speeds (circles) at ultrasonic frequencies on an acrylate plastic block. Inset, a theoretical Rayleigh surface wave pattern. The measured surface displacement at 1 MHz is displayed. **b** Measured asymmetric (A0) and symmetric (S0) Lamb wave modes on a 1 mm thick polystyrene bottom plate (A0: magenta circles, S0: blue circles) and the fitting result (dashed lines). Wavenumber domain plot displayed at 787.6 kHz resolves the two wave modes. **c** Comparison of shear moduli measured by MHz OCE and literature values obtained from static mechanical testing. **d** Measured wave speed on the cortical surface of a bovine tibia bone (circles) and the fitting result (dashed line). In Fig. 4a–d, n = 4 measurements were performed on four different locations of one sample. Data are represented as mean values ± SD.

## Dynamic shear analysis of uniform soft materials

Next, we measured soft polymer samples using surface waves excitation in a frequency range of 0.1 kHz–100 kHz. Figure 5 displays the experimental and analysis data from a rubber (Ecoflex 00–50), poly-dimethylsiloxane (PDMS), and a tough hydrogel[46], respectively. Figure 5a, d, g show measured surface displacement profiles at 40 kHz. The profiles contain not only the Rayleigh waves but also supershear surface waves[47,48], which is evidenced by the two separated peaks in the wavenumber domain (Supplementary Fig. S7). We isolated the Rayleigh wave and derived the complex wave number $k_r + ik_i$ of the bulk shear wave (Supplementary Fig. S8). The measured speed ($v = \omega_m/k_r$) and the coefficient of attenuation ($k_i$) are shown in Fig. 5b, e, h. The complex shear modulus, $\mu' + i\mu''$, is related to the complex wave numbers via:

$$\mu' = \rho\omega_m^2\left(k_r^2 - k_i^2\right)\left(k_r^2 + k_i^2\right)^{-2} \tag{1}$$

$$\mu'' = 2\rho\omega_m^2 k_r k_i\left(k_r^2 + k_i^2\right)^{-2} \tag{2}$$

The measured storage and loss shear moduli are plotted in Fig. 5c, f, i. They show characteristic frequency dependencies due to viscoelasticity, which fit well into a power law model with a single or two slopes across the frequency range. For the rubber, $\mu'$ and $\mu''$ increase from 50 to 160 kPa and from 9 to 323 kPa, respectively, over 0.2 kHz–50 kHz. $\mu''$ grows at a faster rate above 10 kHz and becomes greater than $\mu'$ above 40 kHz. The crossover of $\mu'$ and $\mu''$ causes strong wave attenuation ($k_i/k_r = 0.41$). When $k_i/k_r > 1$, waves become over-damped, and their velocities cannot be determined. Our data agreed with previous measurements over 0.2–7.8 kHz by MRI elastography[49], but high-frequency data above 10 kHz could not be found. For the PDMS, our data up to 130 kHz show no crossover between $\mu'$ and $\mu''$. the low frequency data agree with previously reported storage modulus values of 1.0–2.6 MPa and loss tangents of 0.2–1.0 MPa[50,51], which were measured by time-temperature superposition using DMA below 100 Hz[52]. The tough hydrogel sample showed the lowest viscosity among the three materials. The low viscosity is due to the water that lubricates highly-entangled polymer chains and reduces energy dissipation[53], as previously suggested from DMA results up to 100 Hz[46]. Our data reveals the unusually low $\mu''$ over the entire acoustic range, suggesting that this material is good at transmitting sounds and even ultrasounds.

## Depth-resolved stiffness mapping of cartilage

We applied OCE to bovine cartilage in the knee joint ex vivo (Fig. 6a). The firm, connective tissue consists of three layers with different mechanical properties: the noncalcified cartilage (NCC), calcified cartilage (CC) and the bone (Fig. 6b). In our three-layer model, the thicknesses and shear moduli of NCC, CC and bone layer are denoted by $h_i$ and $\mu_i$ (i = 1, 2, 3), respectively. The NCC layer is readily distinguishable in the OCT image, from which we determined $h_1 = 300$ μm. Unfortunately, the lower part of the CC layer and the underlying bone are not visible due to the limited optical penetration depth. We estimated the thickness of the CC layer to be $h_2 = 2.5$ mm from cross-sectional cuts of equivalent samples (see Methods) and assumed $h_3 \gg h_2$. We measured the elastic wave motions over 10–500 kHz. Figure 6c shows the dispersion of the Rayleigh waves, which shows several features. Besides a dramatic transition between 20 and 40 kHz, there appears to be a weak transition between 80 and 100 kHz. The Rayleigh surface wave has a 1/e amplitude decay depth equal to, approximately, a half wavelength. Therefore, as the frequency increases, the wave is increasingly confined near the surface, and its speed reflects the average stiffness of the region. The high speed below 20 kHz is due to the hard bone ($\mu_3 = 3.27$ GPa, see Fig. 6d). The first downward transition from 20 to 40 kHz is related to the bone-CC interface as the Rayleigh wave moves away from the bone. And the downward transition from 80 to 100 kHz is attributed to the CC-NCC interface and indicates that $\mu_1 < \mu_2$. Using a three-layer theoretical model[54], we fitted the experimental data and plotted the best-fit result in Fig. 6c, from which we determined $\mu_1 = 5.6 \pm 0.2$ MPa and $\mu_2 = 13.2 \pm 0.9$ MPa. These shear elastic moduli are within the range of previously reported values between ~2.3 MPa by MRI elastography[55] and 27.4 MPa by laser scanning vibrometer[56].

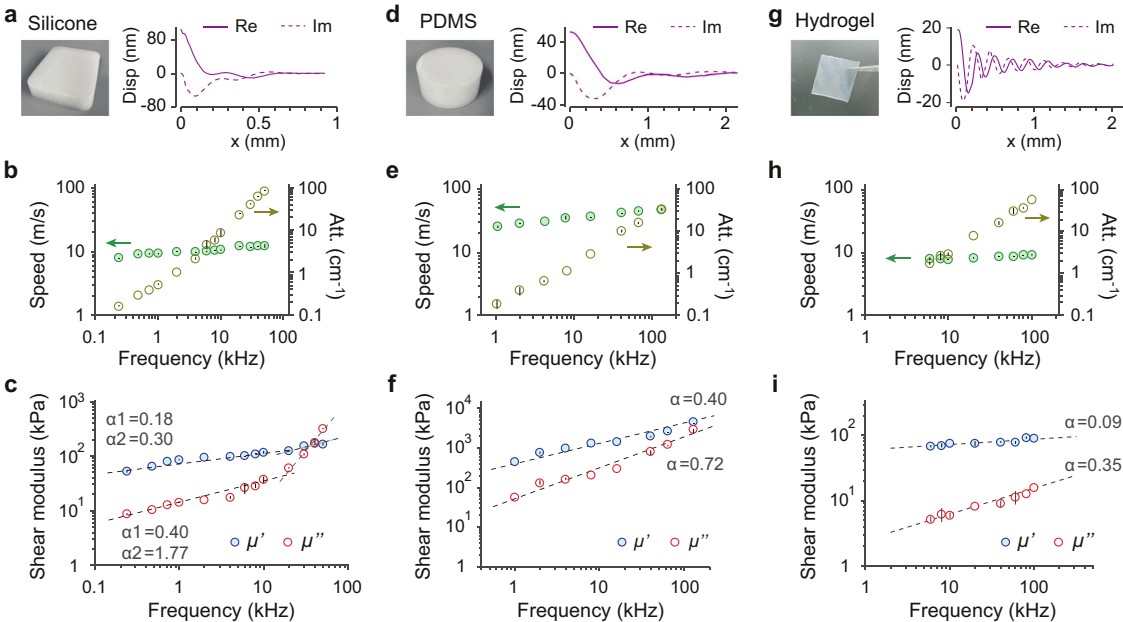

**Fig. 5 | Ultra-wideband shear rheological analysis of uniform soft materials.** **a–c** Rubber, **d–f** PDMS, **g–i** Hydrogel. **a** Picture of the rubber and its surface wave displacement profile at 40 kHz (Re: real part, Im: imaginary part). **b** Rubber's wave dispersion relation and attenuation. **c** Rubber's storage modulus ($\mu'$) and loss modulus ($\mu''$). **d** Picture of the PDMS and its Rayleigh surface wave displacement profile at 40 kHz. **e** PDMS's wave dispersion relation and attenuation. **f** PDMS's storage modulus ($\mu'$) and loss modulus ($\mu''$). **g** Picture of the hydrogel and its Rayleigh surface wave displacement profile at 40 kHz. **h** Hydrogel's wave dispersion relation and attenuation. **i** Hydrogel's storage modulus ($\mu'$) and loss modulus ($\mu''$). Dashed lines in (**c**, **f**, **i**) are curve fits using a power law model: $\mu \propto f^{\alpha}$. For the rubber, a bi-linear fit was used for 0.2–20 kHz and 20–50 kHz separately. In Fig. 5b, c, e, f, h, and i, $n = 3$ measurements were performed on three different locations of one sample. Data are represented as mean values ± SD.

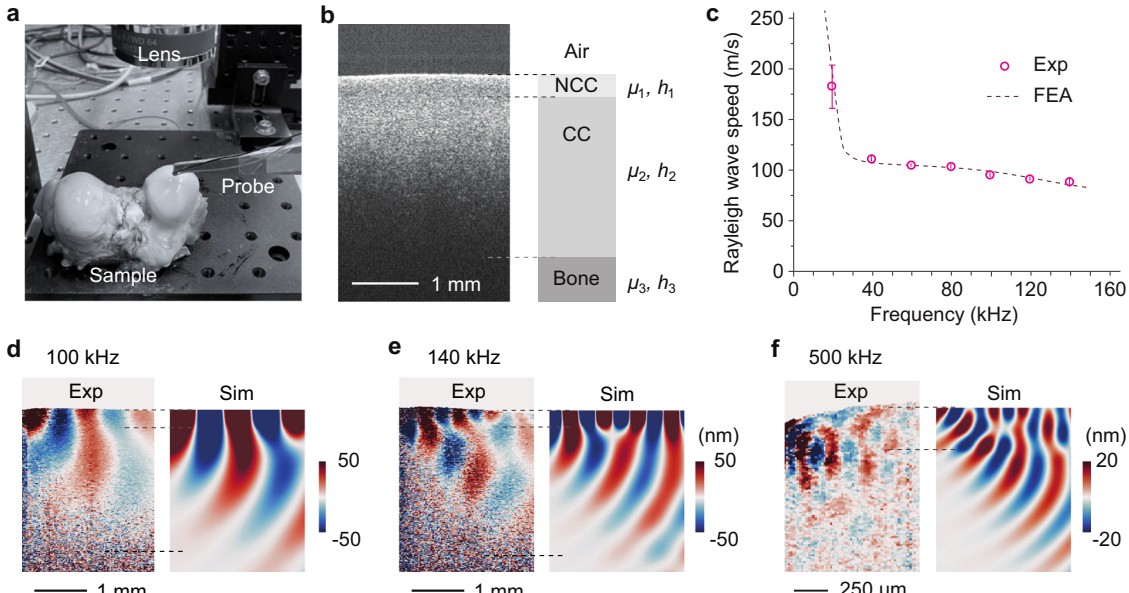

**Fig. 6 | Depth-resolved stiffness measurement of joint tissue ex vivo. a** Picture of a bovine joint sample and the PZT contact probe. **b** Cross-sectional OCT image with three layers labeled: noncalcified cartilage (NCC), calcified cartilage (CC), and the bone. The shear moduli and thickness of the NCC, CC and bone layers are denoted by $\mu_i$ and $h_i$ ($i = 1, 2, 3$), respectively. **c** Measured Rayleigh surface wave speeds (circles) and the best-fit finite element analysis (FEA) result using the three-layer model (dashed curve). Data represent the mean values ± SD of four measurements from two samples at two locations. **d–f** Side by side comparison of the measured cross-sectional displacement image and the FEA simulation at 100 kHz, 140 kHz, and 500 kHz, respectively. Dashed lines indicate the approximate boundaries between the NCC, CC, and bone.

Our demodulation method allowed us to generate subsurface wave maps (Supplementary Fig. S9). Figure 6d shows cross-sectional displacement images obtained at 100, 140, and 500 kHz, respectively. It is apparent that the wavelength is longer in NCC than CC. We performed finite element analysis (FEA) using the geometrical and elastic parameters obtained from the curve fitting to generate displacement maps. The simulation reproduces the measured displacement fields quite well. The optical SNR from the deep CC region is low,

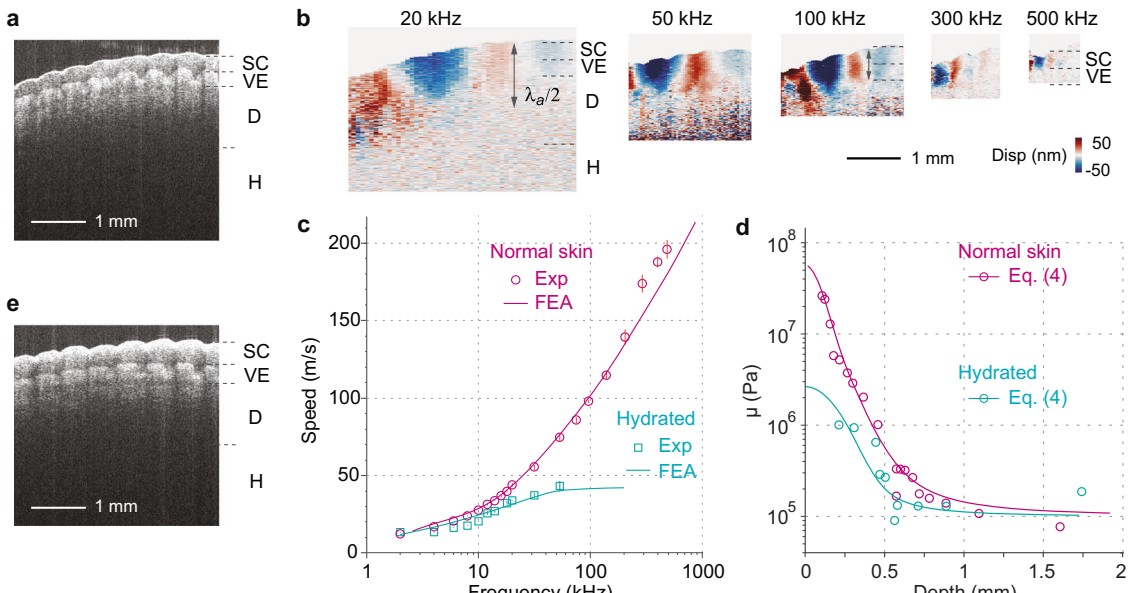

**Fig. 7 | Depth-resolved stiffness mapping of human skin in vivo. a** Cross-sectional OCT image of the fingertip skin in the normal condition. The stratum corneum (SC), viable epidermis (VE), dermis (D), and hypodermis (H) layers are labeled. **b** Cross-sectional displacement images (only the real part is shown) of normal fingertip at different frequencies. The layer interfaces and the half wavelength of the Rayleigh surface wave are marked. $\lambda_a/2$ is 1.1 mm at 20 kHz and 0.2 mm at 500 kHz. **c** Measured Rayleigh surface wave speed over 2–500 kHz and the theoretical fitting curves for normal and hydration conditions. Data are represented as mean values ± SD of $n = 4$ measurements from approximately the same tissue regions (circles). The finite element analysis (FEA) data are also shown (solid lines). **d** Calculated depth profiles of shear modulus for the normal and hydrated tissues. The profiles were obtained using $v \approx 9.5 + \frac{af}{\sqrt{b^2f^2+1}} + \frac{cf}{df+1}$ (m/s), where $a = 0.44$, $b = 0.0032$, $c = 1.6$ and $d = 0.02$ for normal skin, and $a = 1.25$, $b = 0.032$ and $c = 0$ for hydrated skin. **e** Cross-sectional OCT image of the same region after immersing in warm water for 20 min.

and light penetrates poorly into bones because of strong light scattering. Nonetheless, substantial energy of elastic waves reaches the bone, partially reflected from and partially transmitting into the bone. This boundary effect of elastic waves affects the wavelength of the wave along the surface and is reflected on the wave dispersion curve.

## Depth-resolved stiffness mapping of in vivo human skin

We next examined the stiffness of human fingertip skin in vivo over 100 Hz–500 kHz. We have previously measured the moduli of the epidermis, dermis, and hypodermis in the forearm skin by OCE at acoustic frequencies[33]. With ultrasonic frequencies, we were interested in resolving the stratum corneum[57] and the viable epidermis layers within the epidermis (Fig. 7a). Figure 7b show cross-sectional displacement images at different frequencies. At 20 kHz, the Rayleigh wave is confined predominantly in the SC and VE layers. Above 300 kHz, the majority of the elastic energy resides in the SC layer.

The wave velocity dispersion curve has a strong frequency dependence in the normal skin condition (Fig. 7c, red circles). The wave speed increases dramatically with frequency, an opposite trend to the cartilage. This is consistent with the elastic modulus profile of the skin, which tends to decrease with depth. As the frequency increases, the surface elastic waves are increasingly more localized in the upper stiffer region (see Supplementary Fig. S10). Because the elastic modulus of epidermis is greater than dermis and hypodermis, the surface waves are leaky waves, and this anti-waveguiding loss makes it difficult to extract the intrinsic material absorption from measured wave profiles. Therefore, we approximated the skin layers as pure elastic materials. To relate the dispersion to shear modulus, let us consider a depth-profile of elastic shear modulus, $\mu(z)$. There are no analytic solutions for wave propagation for arbitrary nonuniform modulus. For small nonuniformity, the wavenumber can be estimated from $k^2 = \rho\omega^2 \int_0^\infty \psi^* \frac{1}{\mu(z)} \psi dz$, where $\psi(z)$ is the wave function in a uniform medium with a space-averaged modulus: $|\psi(z)|^2 \approx 2k_z e^{-2k_z z}$, where $k_z \approx 0.31 k$. We may extend the perturbation theory to large

variation with a simplified form:

$$k^2 \approx \frac{\rho\omega^2}{L_z} \int_0^{L_z} \frac{1}{\mu(z)} dz \quad (3)$$

where $L_z$ denotes the penetration depth of the Rayleigh wave. We may choose $L_z = 1/2k_z \approx 0.25\,\lambda$ or more generally, $L_z = a\lambda = av/f$, where $a$ is a fitting constant. By differentiating both sides of Eq. (3) with respect to $f$, we obtain

$$\mu(z) \approx \rho v^2 (v - fv')/(v + fv') \quad (4)$$

where $z = av/f$, $v = 1.048\,v_R$ is bulk shear velocity obtained from the measured Rayleigh wave velocity $v_R(f)$, and $v' = dv/df$. Notice that in bulk isotropic materials, where $v' = 0$, Eq. (4) is reduced to $\mu = \rho v^2$. Figure 7d shows $\mu(z)$ obtained from the experimental data. Also shown are depth profiles calculated using Eq. (4) using an empirical best-fit function to the data in Fig. 7c. We find $\mu(z)$ to decay rapidly from 56 MPa at the surface to 100 kPa at the asymptotic depth (using $a = 0.25$). This finding is consistent with the fact that water content decreases exponentially from the VE to the surface of SC[58] and that the elastic modulus decreases exponentially with depth in SC.

We measured the same skin immediately after the fingertip was immersed in warm water for 20 min. The thickness of the SC layer increased from 0.31 mm to 0.41 mm due to swelling (Fig. 7e). The wave velocity at high frequencies decreased significantly (Fig. 7c, green squares), indicating a softening of the SC layer. The Rayleigh waves became overdamped at frequencies above 50 kHz due to high viscous damping. Using Eq. (4), we obtained $\mu = 2.6$ MPa at the surface of the hydrated SC.

To verify, we performed FEA simulation by modeling the skin as a four-layer structure. We measured the shear moduli of the dermis ($\mu_2 = 9$–17 kPa) and hypodermis ($\mu_3 = 2$–3 kPa) layers from wave dispersion below 1 kHz[33]. The shear modulus in the SC layer at the pre-

hydrated condition was assumed to have an exponentially decreasing function from $\mu_0 = 50$ MPa at the surface to $\mu_1 = 2$ MPa at the interface between the SC and VE layers (Supplementary Fig. S11). In the hydrated condition, the stiffness of the swollen SC layer was assumed to be uniform and the same as the VE layer ($\mu_0 = \mu_1 = 2$ MPa). In Fig. 7c we plot the dispersion relations obtained by the FEA simulation, which captures the salient features of the dispersion relations.

## Discussion

We have demonstrated an OCE system capable of visualizing elastic waves within various materials and tissues over a wide range of shear modulus from 1 kPa to 10 GPa and across an unprecedented frequency range from static to a few MHz. The fundamental reflectivity-limited performance of the system provided sufficient SNR to determine wave velocities with high accuracy. The measured speed dispersion over the wide frequency range was critical to extract depth-dependent stiffness from multi-layered tissues commonly found in various tissues. The contact area of the PZT probe was optimized to be less than half the wavelength to maximize acoustic impedance matching from the actuator motion to the elastic waves. Throughout the experiments, the peak wave amplitude ranged from several hundreds of nm at acoustic frequencies to as small as 10 nm at ultrasonic frequencies.

The demodulation scheme was highly effective and allowed the upper frequency limit to be no longer limited by the A-line rate. This relaxes the system requirement. The demodulation scheme should be applicable to any swept source OCT systems regardless of their A-line rates. Ultra-high speed OCT systems with ultrahigh A-line rate up to several MHz are available using laser techniques such as Fourier domain mode locking[35,36] and stretched-pulse active mode locking[37], and they will be well suited to build into MHz-range OCE systems without aliasing. In both aliased and non-aliased schemes, all the acquired data points are used for signal processing. Therefore, in principle the same SNR is expected for the same data acquisition time, and we expect shot noise-limited performance in both schemes as long as practical noises, such as the phase noise of laser sweep, are minimized. Ultimately, the maximum frequency for the wave based OCE technique is limited by viscous damping in samples, which can make waves overdamped. The critical frequency seems to be at a few hundreds of kHz for soft materials.

It should be noted that the aliasing scheme does not work for spectra-domain OCT because signal modulation during the A-line acquisition period is averaged out due to the integrating nature of CCD spectrometers (equivalent detector bandwidth is only $f_A/2$). This is analogous to the phase washout motion artifact in spectral-domain OCT[34]. Swept-source OCT is free from the phase washout as long as $f_m$ is lower than the detector bandwidth, which is typically 1000 times $f_A$.

We envision ultra-wideband OCE to be a powerful tool for mechanical characterizations of soft and hard tissues with high spatiotemporal resolution. Aside from mapping stiffness variation along the depth in sample, the OCE technique can also be used to evaluate stiffness variations along the lateral plane with spatial resolutions in the order of a couple of elastic wavelengths. With improved algorithm, it should be possible to reveal 3-dimensional mechanical heterogeneity of sample from wide-band dispersion curves. In terms of applications, ultra-wideband OCE may be useful in capturing the full spectrum of the complex rheology of tissues[59–61] and biomaterials[62,63]. Ultra-wideband OCE has the potential to enable novel biomechanics-based medical diagnostics[64–66]. For example, the ability to detect up to MHz excitation may be useful to study fast neuronal activities through neuro-mechanical coupling and develop novel therapeutics to treat brain diseases[67]. Furthermore, ultra-wideband OCE may be used to extract mechanical stress in thin structures without prior

knowledge of material properties[68]. Finally, the emergence of deep learning offers a new approach to apply the information-rich elastography data to multi-parameter inverse problems[69].

## Methods

### Swept-source OCT signal

The photodetector current $I(t)$ of the OCT interferometry from a point scatter located at $z_0$ can be expressed as

$$I(t) = rP(t)\cos\left(2k_0z_0 + 2k_1tz_0\right),\qquad(5)$$

where $r$ denotes the reflectance amplitude, $P(t)$ denotes the optical power profile, $k = k_0 + k_1 t$ denotes the wavenumber of the swept source. The Fourier transformation of the photocurrent acquired during time interval $-T/2 \leq t \leq T/2$ with respect to $\hat{k} = 2k_1 t$, denoted by $F(z)$, is given by

$$F(z_0;z) = \int_{-k_1T}^{k_1T} I(\hat{k})e^{i\hat{k}z}\,\mathrm{d}\hat{k},\qquad(6)$$

where $T$ is the period of wavelength sweep (A line period). We assume that the optical power profile has a Gaussian shape with a full width at half maximum (FWHM) $\sigma T$, i.e., $P(t) = \exp\left[-4\ln2\frac{t^2}{(\sigma T)^2}\right]$. Then we get

$$F(z_0;z) = \frac{r}{2}\int_{-k_1T}^{k_1T} \exp\left[-4\ln2\frac{\hat{k}^2}{(2\sigma k_1 T)^2}\right]\left\{e^{-i\left(2k_0z_0+\hat{k}z_0\right)} + \text{c.c.}\right\}e^{i\hat{k}z}\,\mathrm{d}\hat{k},\qquad(7)$$

where 'c.c.' denotes the complex conjugate. When $\sigma < 1$, we can approximate the integral range to infinity and get

$$F(z_0;z) = \frac{r}{2}\left(\sqrt{\frac{\pi}{\ln2}}\sigma k_1 T\right)\{F_0(z) + \text{c.c.}\}.\qquad(8)$$

where we introduce $F_0(z)$ that is defined as

$$F_0(z) = e^{-4\ln2\frac{(z-z_0)^2}{(\sigma_z)^2}}e^{-i2k_0z_0}.\qquad(9)$$

where $\sigma_z \equiv \frac{4\ln2}{\sigma k_1 T}$ corresponds to the FWHM axial resolution.

Without loss of generality, the displacement of the scatter along the laser beam can be expressed as $\Delta z(t) = \delta g(t)$, where $\delta$ is a constant and $g(t)$ is a dimensionless function of time. Then the location of the scatter is $z_0 + \Delta z$. Since the laser is periodically tuned, we introduce $\hat{t} = t - mT$ for the $m$-th A line acquired during $mT - T/2 \leq t \leq mT + T/2$, and $g_m(\hat{t}) = g(\hat{t} + mT)$. Substitution of $z_0$ and $P(t)$ in Eq. (5) with $z_0 + \delta g_m(\hat{t})$ and $P(\hat{t})$, we can get

$$I_m(\hat{t}) = rP(\hat{t})\cos\left(2k_0z_0 + 2k_1\hat{t}z_0 + 2k_0A_{z_0}g_m(\hat{t}) + 2k_1\hat{t}\delta g_m(\hat{t})\right).\quad(10)$$

We make assumption that the wavelength tuning range is a small fraction of the mean wavelength, $k_1\hat{t} \ll k_0$, to get

$$\begin{aligned}
I_m(\hat{t}) &\approx rP(\hat{t})\cos(2k_0z_0 + 2k_1\hat{t}z_0 + 2k_0\delta g_m(\hat{t}))\\
&= rP(\hat{t})\left\{e^{-i(2k_0z_0 + 2k_1\hat{t}z_0 + 2k_0\delta g_m(\hat{t}))} + \text{c.c.}\right\}\\
&= rP(\hat{t})\left\{e^{-i(2k_0z_0 + 2k_1\hat{t}z_0)}\left[1 + (-2ik_0\delta)g_m(\hat{t}) + \frac{1}{2}(-2ik_0\delta)^2 g_m^2(\hat{t}) + \dots\right] + \text{c.c.}\right\}.
\end{aligned}$$
$$(11)$$

We define

$$G_m^{(1)}(z) = \int_-^+ g_m(\hat{t}) e^{i\hat{k}z} \, d\hat{k},$$

$$G_m^{(2)}(z) = \int_-^+ g_m^2(\hat{t}) e^{i\hat{k}z} \, d\hat{k}. \tag{12}$$

The Fourier transform of the photocurrent $I(\hat{t})$ is

$$F_m(z_0;z) \approx \frac{r}{2} \left( \sqrt{\frac{\pi}{\ln2}} \sigma k_1 T \right) \left\{ \left[ F_0(z) + (-2ik_0\delta) F_0(z)*G_m^{(1)}(z) \right. \right.$$
$$\left. \left. + \frac{1}{2}(-2ik_0\delta)^2 F_0(z)*G_m^{(1)}(z) + \ldots \right] + \text{c.c.} \right\}, \tag{13}$$

where '*' denotes the convolution.

For harmonic vibration studied here, $\Delta z = \delta \sin(\omega_m t + \varphi)$, where $\delta$, $\varphi$, and $\omega_m$ denote the vibration amplitude, initial phase, and angular frequency, respectively. Then have $g_m(\hat{t}) = \sin(\omega_m \hat{t} + m\omega_m T + \varphi)$, and

$$G_m^{(1)}(z) = \frac{1}{2i} \left[ e^{i(m\omega_m T + \varphi)} \delta\left(z + \frac{\omega_m}{2k_1}\right) - e^{-i(m\omega_m T + \varphi)} \delta\left(z - \frac{\omega_m}{2k_1}\right) \right],$$

$$G_m^{(2)}(z) = \frac{1}{2} \left[ \delta(z) - \frac{1}{2}\left( e^{2i(m\omega_m T + \varphi)} \delta\left(z + \frac{\omega_m}{k_1}\right) + e^{-2i(m\omega_m T + \varphi)} \delta\left(z - \frac{\omega_m}{k_1}\right) \right) \right], \tag{14}$$

where $\delta(z)$ is the Dirac delta function. Inserting Eq. (13) we get

$$F_m(z_0;z) \approx \frac{r}{2} \left( \sqrt{\frac{\pi}{\ln2}} \sigma k_1 T \right) \left\{ \left[ (1 - (k_0\delta)^2) F_0(z) - (k_0\delta)\left( e^{i(m\omega_m T + \varphi)} F_0\left(z + \frac{\omega_m}{2k_1}\right) \right.\right.\right.$$
$$\left.\left. - e^{-i(m\omega_m T + \varphi)} F_0\left(z - \frac{\omega_m}{2k_1}\right) \right) + \frac{1}{2}(k_0\delta)^2\left( e^{2i(m\omega_m T + \varphi)} F_0\left(z + \frac{\omega_m}{k_1}\right) \right.\right.$$
$$\left.\left.\left. + e^{-2i(m\omega_m T + \varphi)} F_0\left(z - \frac{\omega_m}{k_1}\right) \right) + \ldots \right] + \text{c.c.} \right\}. \tag{15}$$

Taking the first order approximation, we have

$$F_m(z_0;z) \approx \frac{r}{2} \left( \sqrt{\frac{\pi}{\ln2}} \sigma k_1 T \right)$$
$$\left\{ \left[ F_0(z) - (k_0\delta)\left( e^{i(m\omega_m T + \varphi)} F_0\left(z + \frac{\omega_m}{2k_1}\right) - e^{-i(m\omega_m T + \varphi)} F_0\left(z - \frac{\omega_m}{2k_1}\right) \right) \right] + \text{c.c.} \right\}. \tag{16}$$

## Aliasing of the vibration signal

To derive the vibration signal from the OCT measurement, we consider one of the side lobes in Eq. (16), $F_m\left(z_0; z_0 - \frac{\omega_m}{2k_1}\right)$, and find

$$F_m\left(z_0; z_0 - \frac{\omega_m}{2k_1}\right) \propto (k_0\delta) e^{-i2k_0 z_0} e^{im\omega_m T} e^{i\varphi}, \tag{17}$$

which retains the vibration signal. Denote the frequency $f_m = \frac{\omega_m}{2\pi}$ and the A line rate $f_A = \frac{1}{T}$. When $f_m > 0.5 f_A$, Eq. (17) gives rise to the aliasing effect. Denote $f_m = \bar{f}_m + 0.5n f_A$, where $n$ is an integer to make $0 \le \bar{f}_m < 0.5 f_A$. Then,

$$F_m\left(z_0; z_0 - \frac{\omega_m}{2k_1}\right) \propto (k_0\delta) e^{-i2k_0 z_0} e^{i2\pi m \bar{f}_m T} e^{i\varphi}, \tag{18}$$

when $n$ is an even number, and we can get the apparent frequency $\bar{f}_m$.

$$F_m\left(z_0; z_0 - \frac{\omega_m}{2k_1}\right) \propto (k_0\delta) e^{-i2k_0 z_0} e^{i2\pi m(\bar{f}_m - 0.5 f_A)T} e^{i\varphi}, \tag{19}$$

when $n$ is an odd number and we get the apparent frequency $(\bar{f}_m - 0.5 f_A)$.

## Demodulation of vibrations from internal scatters

Given the axial coordinate $z_0$, without loss of generality, the Fourier transform of the photocurrent for the $m$-th scan, denoted by $\mathcal{F}_m(z_0)$, is

$$\mathcal{F}_m(z_0) \approx F_m(z_0;z_0) + F_m(z_0 + z_m;z_0) + F_m(z_0 - z_m;z_0), \tag{20}$$

Where $z_m = \frac{\omega_m}{2k_1}$. The latter two, comes from the contributions of the sidelobes of two scatters located at $z_0 + z_m$ and $z_0 - z_m$ (denoted by S' and S''), respectively. As the sidelobes contain the vibration signals, we expect to extract the vibrations of S' and S'' from $\mathcal{F}_m(z)$.

Note that the sum $\sum_{m=1}^N F_m(z_0 \pm z_m;z_0) = 0$, given that $Nf_m/f_A$ is an integer, where $N$ is the number of A lines. We have $\frac{1}{N}\sum_{m=1}^N \mathcal{F}_m(z_0) = F_m(z_0;z_0)$, the static OCT signal of the scatter located at $z_0$. Subtracting this term from $\mathcal{F}_m(z_0)$ we can get,

$$F_m(z_0 + z_m;z_0) + F_m(z_0 - z_m;z_0) \approx \mathcal{F}_m(z_0) - \frac{1}{N}\sum_{m=1}^N \mathcal{F}_m(z_0). \tag{21}$$

The left side of Eq. (21) contains the left sidelobe of the scatter S' and the right sidelobe of scatter S''. Here we show the extraction of the vibration for S' as an example. Note that $\frac{1}{N}\sum_{m=1}^N F_m(z_0 + z_m) = F_m(z_0 + z_m; z_0 + z_m)$, the static OCT signal of S'.

$$\left(k_0 \delta_{z_0 + z_m}\right) e^{i(m\omega_m T + \varphi)} + \frac{F_m(z_0 - z_m;z_0)}{\frac{1}{m}\sum_{m=1}^M \mathcal{F}_m(z_0 + z_m)} \approx \frac{\mathcal{F}_m(z_0) - \frac{1}{N}\sum_{m=1}^N \mathcal{F}_m(z_0)}{\frac{1}{N}\sum_{m=1}^N \mathcal{F}_m(z_0 + z_m)}. \tag{22}$$

where $\delta_{z_0 + z_m}$ denotes the vibration amplitude at $z_0 + z_m$. The first term of the left side in Eq. (22) gives the vibration of S'. To eliminate the second term that results from the contribution of the right sidelobe of S'', we note that the phase change of the right sidelobe has an opposite direction as the left side lobe. The two terms can be separated in the frequency domain. Then the last step to extract the vibration of S' is to perform Fourier transform on the right side of Eq. (22).

## Experimental setup

Our experimental setup (Supplementary Fig. S1) is based on a home-built, swept source OCT system[70]. The system uses a rotating-polygon-mirror wavelength-swept laser[71] with a central wavelength of 1307 nm and a 3-dB bandwidth of 80 nm at an A-line rate of 43.2 kHz. The axial resolution is 16 μm. The optical beam is scanned by a pair of galvanometer mirror scanners and focused by a wide-aperture scan lens (Thorlabs, LSM54−1310) yielding a long working distance of 64 mm and a transverse resolution of ~30 μm. The average optical power on the sample is 12 mW. A fiber Bragg grating (FBG) and photodiode (PD) provide a pulse signal synched to each wavelength sweep cycle of the laser output. This FBG optical clock ensures time synchronization among the modulation waveform to PZT, OCT beam position scan, and OCT data acquisition. The signal from a dual-balanced detector (Thorlabs, PDB110C, 100 MHz) was digitized by a data acquisition board (Signatec, PX14400, 14 bit) at a sampling rate of 108 MHz. A total 2048 data points were acquired during each A-line. An input/output (I/O) board (National Instruments, USB-6353) is used to generate analog waveforms for the galvanometer scanners. The I/O board or a function generator (Tektronix, AFG3021C) is used to generate stimulus waveforms. The waveforms are smoothed with a reconstruction filter (Thorlabs, EF122) before feeding to the PZT (Thorlabs, PA4CEW). During data acquisition, the stimulus waveforms applied to the PZT are simultaneously recorded by the I/O board, which was later used for time jitter correction. Figure S3 depicts the timing diagram for the data acquisition protocol. The M-B scan protocol comprises the acquisition of $N$ consecutive A-lines at each transverse location. The OCT

beam is then moved to the next position, and the stimulus waveform is repeated. A total of 96 positions are acquired. Typical, we set $N$ to be in the range of 100–250. For $N = 108$, a single M-scan at each transverse location takes 2.5 ms, and the total measurement time per frequency is 0.24 s.

The harmonic waveforms are sent to a wideband PZT via an amplifier (PiezoDrive, PDm200B for frequencies below 200 kHz, and E&I 1020L above 200 kHz). For the ultrasonic OCE of hard materials, the PZT was in direct contact with the sample with a contact area of 2 mm × 2 mm to generate sufficient push force. For the dynamic shear analysis of soft uniform materials, a custom mechanical actuator comprising of a PZT and a 3D-printed cylindrical tip with 0.6 mm radius was used. The tip was designed to facilitate the theoretical model of near-field surface wave[48]. For the depth-resolved stiffness mapping of the joint and skin, a 3D-printed prism-shape tip with a line contact of 2 mm is used to reduce wave attenuation and suppress supershear surface wave[33].

### Displacement field analysis

All algorithms were developed in MATLAB (MathWorks Inc., MATLAB 2019). The raw data collected from the M-B scan was processed using the standard swept-source phase-stabilized algorithm to obtain the complex-valued OCT tomogram[70]. When $f_m > f_A/2$, the correct vibrations occurred at two side lobes. We typically selected the left side lobe, and then applied the demodulated algorithm to extract the surface displacement profiles. Next, we performed a 1-dimensional Fourier transform to move the data from time $t$ domain to frequency $f$ domain. The frequency domain data was filtered at the driven frequency (when $f_m < f_A/2$) or the aliased frequency (when $f_m > f_A/2$) to obtain lower noise waveforms. After we obtained the displacement profiles over the $x$ coordinate, we performed another 1-dimensional Fourier transform to move the data from the spatial $x$ domain to the wavenumber $k$ domain. The wavenumber $k$ of the surface wave was then determined from the plot by selecting the peak corresponding to the Rayleigh surface wave. This filtering in the $k$-$x$ domain is critical to remove other higher-order modes especially at high frequencies. The phase velocity is then given by $v = 2\pi/k$. To obtain the cross-sectional displacement image, the same method used for obtaining the surface wave displacement was applied throughout the entire pixels along the depth $z$. In addition, the motion artifact caused by the surface wave motion were corrected within the sample[72].

### Laser Doppler vibrometer experiment

We measured the vibration amplitude on a PZT using a laser Doppler vibrometer (Polytec, HLV 1000). The experimental setup is shown in Fig. S3. The drive waveform was generated by a National Instrument board and amplified by a power amplifier (Crest Audio, 1001A) before feeding to the PZT. The stimulus waveform and data acquisition are time synchronized. A retro-reflective adhesive tape was adhered to the surface of the PZT to enhance the vibrometer signal. The laser Doppler vibrometer can reliably measure vibration frequencies up to 80 kHz. The stimulus frequency was varied from 50 to 80 kHz with a 10 kHz increment. Data were averaged 100 times to improve SNR. After finishing the laser vibrometer experiment, the same PZT was measured under the OCE setup. The linear slope efficiency of the PZT at 80 kHz was measured for a voltage range from 0 to 5 V. The measured slope efficiencies (in nm/V) were compared between the vibrometer and the OCE system.

### Measure complex wave number from surface waves

For soft materials, the near-field displacement is primarily dominated by the Rayleigh and supershear surface waves in high frequency regime. An analytical approximation solution for the surface waves excited by a cylindrical probe can be obtained. The surface displacement at lateral coordinate $x$ is[48]

$$u(x) = i\pi \frac{a p_0 K^4}{\rho \omega_m^2} \left[ \frac{J_1(K_R a) K_R}{\mathcal{F}'(K_R)} H_0^{(1)}(-K_R x) + \frac{J_1(K_{SS} a) K_{SS}}{\mathcal{F}'(K_{SS})} H_0^{(1)}(-K_{SS} x) \right],$$

(23)

where $J_1$ is the Bessel function of the first kind of order 1, $H_0^{(1)}$ is the Hankel function of the first kind, $a$ is the radius of the cylindrical probe, $p_0$ is the pressure amplitude exerted to the sample by harmonic vibrations of the probe. $\omega_m$ is the vibration frequency. $K$, $K_R$ and $K_{SS}$ are wavenumbers of the shear, Rayleigh surface, and supershear surface waves, respectively. $K_R$ and $K_{SS}$ are roots of the secular equation $\mathcal{F}(\kappa) = (2\kappa^2 - K^2)^2 - 4\kappa^2 \sqrt{\kappa^2(\kappa^2 - K^2)} \text{sign}\{\text{Re}(\kappa^2 - K^2)\} = 0$. So, we have $K_R = 1.047 K$ and $K_{SS} = (0.4696 - 0.1355i)K$. $\mathcal{F}'(\kappa) = d\mathcal{F}/d\kappa$. To derive $K$ from the experiments, we fit the real and imaginary displacements simultaneously with Eq. (23) using the least-squares method. Representative fitting results can be found in Supplementary Figure S9.

### Finite element analysis

FEA was performed with Abaqus 6.12 (Dassault Systèmes). We used plane strain models and performed time domain simulations. A localized harmonic surface pressure was applied to excite elastic waves, simulating the PZT actuator in the experiments. Other sides of the model were completely constrained. The geometry and total time of the model was scaled according to the wave wavelength and stimulus frequency, respectively. The time increment and total time were $0.1/f_m$ and $20/f_m$, respectively. The width and height of the model were about 16 folds of the maximum wavelength to avoid reflection from the boundaries. The model was divided into uniform layers. For the simulation of the knee joint, the thickness of each layer was determined from the OCT images. The top layer thickness was 300 μm, the middle layer thickness was 2.5 mm, and the thickness of the bottom layer was set to about 16 times the maximum wavelength (for the lowest frequency) minus the thickness of the first two layers. The shear modulus values of the three layers from top to bottom were 5.6 MPa, 13.2 MPa, and 3.27 GPa, respectively. The Poisson's ratios for the top, middle, and bottom layers were 0.4999, 0.499, and 0.3, respectively. A gradient mesh was adopted to reduce computational costs. At the surface the mesh size was smaller than one tenth of the wavelength and at the bottom the mesh size was about half of the wavelength. The element type used in this study was 8-node biquadratic element (CPE8RH). The convergence of the model was checked by making sure the results independent of the mesh size.

### Preparation of hard materials

The acrylate plastic block (McMaster-Carr) presented in Fig. 4a had a thickness of 5 mm and an area of 50 mm × 50 mm. The polystyrene petri dish (Fisher Scientific) presented in Fig. 4b had a bottom thickness of 1 mm and a diameter of 150 mm. The borosilicate glass coverslip (Fisher Scientific) measured in Supplementary Fig. S6a had a thickness of 0.15 mm and an area of 24 mm × 50 mm. The copper foil (All Foils Inc.) measured in Supplementary Fig. S6b had a thickness of 0.3 mm and an area of 15 mm × 40 mm. The glass coverslip and the copper foil were placed on a lens holder so that the sample was bounded by the air on two sides.

### Preparation of soft materials

The bulk silicone rubber has a diameter and height of 60 mm and 12 mm, respectively. It was prepared from Ecoflex 0050 material (Smooth-On Inc) by mixing the Ecoflex 1A and 1B at 1:1 ratio by weight. The mixture was poured into a mold and cured at room temperature overnight. The material was then post-cured in an oven at 80 °C for 2 h. The bulk PDMS sample has a diameter and height of 120 mm and

50 mm, respectively. It was prepared by using a 2:1 mixing ratio of base elastomer and curing agent (Sylgard 184, Dow Corning) and cured for 45 min at 85 °C. The hydrogel sheet had a size of 40 mm × 40 mm and a thickness of 1.3 mm under fully swollen, highly entangled condition[46]. All soft materials were assumed to have a mass density of $\rho \simeq 1000 \text{ kg/m}^3$.

## Cartilage tissues

Two fresh bovine tibia bones from two juvenile calves were obtained 1 h post-mortem (Research 87 Inc., Boylston, MA). The tissues were wrapped in a wet towel to keep them well-hydrated until use. Three measurements were performed on the cortical surface of the bone. Four measurements were performed on the femoral condyle region covered by articular cartilage. To measure the thickness of calcified cartilage layer, we made a cross-sectional cut after the OCE experiments.

## Human subject

We measured the skin of the left index finger on a 31-year-old male subject. The study was conducted at the Massachusetts General Hospital following approval from the Institutional Review Board of Massachusetts General Hospital and the Mass General Brigham Human Research Office. Written informed consent was obtained from both subjects prior to the measurement. All methods were performed in accordance with the relevant guidelines and regulations. The clinical trial is registered at www.clinicaltrials.gov (National Clinical Trial Identifier: NCT03230981). The measurement site was marked by a surgical grade skin marker. For the hydration test, the whole index finger was immersed in warm water for 20 min, wiped with a clean towel, and then measured immediately.

## Statistics and reproducibility

All the statistical analysis was performed using Microsoft Excel (Microsoft Inc.), and MATLAB R2019a (MathWorks, Inc.) software. All quantitative results were presented as mean ± standard deviation. All the wave speed measurements in the main text and supplementary information have been repeated three times or more independently with similar results.

## Reporting summary

Further information on research design is available in the Nature Portfolio Reporting Summary linked to this article.

## Data availability

The authors declare that all data supporting the findings of this study are available within the article, and the Supplementary Information/Source Data file. Source Data file has been deposited in Figshare under accession code DOI link[73]. Additional data are available from the corresponding authors.

## Code availability

Code for analysis is provided on GitHub under accession code DOI link[74]. Additional codes related to this study are available from the corresponding authors.

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

## Acknowledgements
This study was supported by funding from the National Institutes of Health via grants R01-EY027653 (S.H.Y.), R01-EY033356 (S.H.Y.), and R01-HL098028 (S.H.Y.). The authors thank Dr. Guogao Zhang at Harvard University for providing the hydrogel sample. The authors also thank Dr. Jeffery Tao Cheng at Massachusetts Eye and Ear for helping in the laser vibrometer experiment.

## Author contributions
S.H.Y. conceived the idea. G.Y.L. and S.H.Y. developed the theory and signal processing. X.F. and G.Y.L. carried out the experiments and analyzed the results. All authors wrote and reviewed the manuscript.

## Competing interests
The authors declare no competing interests.
