## [Peer Review file · Nature Communications]

Ultra-wideband optical coherence elastography from acoustic to ultrasonic frequenciesReviewer #1 (Remarks to the Author):

In this work, the authors present an optical coherence elastography technique to extend the frequency range of dynamic elastography from the kHz range to the MHz range. Typically, the frequency range of the elastic wave is considered to be limited by the A-line rate. The technique presented here takes advantage of this property by intentionally causing an aliasing artifact, then performing anti-aliasing demodulation to perform high-frequency excitations effectively. Using this technique, the authors then demonstrate a variety of applications by performing mechanical estimation in hard materials, soft materials, and multilayered tissue, such as cartilage on bone and in vivo human skin.

The study is very interesting, and the technique enables OCE to be used in a broad set of cases. However, there are certain areas where the article could be improved.

1. "...Now that the modulation component is separated from the carrier, the vibration amplitude... manifest itself in the amplitude of the Fourier component..."

This claim is fundamental to the article, as it determines how vibration amplitude is detected in this work, and the method is different from how vibration amplitude is typically detected in phase-sensitive OCE. Later, the authors build on the claim with the following:

"...The side lobes were lower by 20 dB in signal power (10 dB in amplitude) than the main lobe. Since the power ratio is $20 \log_{10}(k_0 \delta)$, we get $\delta = 20.4 \text{ nm}$..."

These claims should be validated, showing that the measured vibration amplitude corresponds to the actual. Validation could be easily performed on a mirror mounted on a PZT.

2. The authors use a histogram to show the noise in the system, then describe a noise compensation technique to reduce the jitter. The authors should show a second histogram representing the difference before and after correction.

3. In Figures 2c and 2d, the x-axis labels seem to be flipped between the two images.

4. "...We obtained the phase of the modulation waveform (applied to the PZT) and subtracted it from the phase of the demodulated signal of each M-scan (Supplementary Note 1)..."

This was not very clear from Supplementary Note 1.

5. In the results, the authors define the following equation, $z_m = \omega_m / (2k_1)$, then later in the methods the equation is defined as $z_m = \omega_m / (2\kappa_1)$, where k was replaced with κ . The authors should go back and ensure that the correct variables are used appropriately.

6. Supplementary Figure 9 should be moved to the main manuscript, as it's the clearest explanation of the author's technique. However, the figure can be improved. It's not immediately clear that Fig 9a) represents three unique signals; it almost seems like the same signal shifted. Additionally, in Figure 9b), there is no explanation in the caption for the gray dotted line in the caption or the figure. Showing the signal background that was subtracted might also be helpful.

7. Relating to the previous point, the organization of the paper could be improved. The first section of the results essentially summarizes the demodulation technique from a theoretical standpoint, which is further expanded in the methods, and the experimental procedure to actually perform the demodulation is relegated to the supplementary materials. Focusing more on the experimental procedure could make the work more impactful and clearer to a wider audience.

8. There are a few frequencies at play in this work, the modulation frequency, the A-line rate, and the carrier frequency (reflector). These are very vaguely defined and designated; correcting this could make the work easier to understand.

9. Why was only the hydrogel data (Figure 4i) curve fitted?

10. The discussion section is quite bare. What are the limitations of the study? Some topics that could be discussed include the following:

High-speed OCT (such as FDML swept source OCT, ~ 3 MHz sweep rate) could theoretically be used to obtain similar results using only the conventional OCE technique. The authors should discuss how the results may compare and the strengths and weaknesses of both approaches. Moreover, high-speed OCT could be used to validate the proposed method at higher frequencies, something that was not done (but could/should be) in this research.

The authors describe in the introduction the problem with wave damping over short distances for high frequencies. Their own results show the same problem (Figure 5 and Figure 6), as the proposed technique can't effectively solve that problem. How might this impact the utility of the proposed technique?

The authors compare their results to the available literature, but the comparison is unavailable for the PDMS, hydrogel, and cartilage.

Reviewer #2 (Remarks to the Author):

Reviewer #2 Attachment on the following page

Major revision

This manuscript demonstrates a novel method to detect high-frequency elastic waves and reconstruct the mechanical properties accordingly. Conventional OCT utilizes the Fourier transform of the spectrum to acquire depth information, in this work, the authors used the frequency of the elastic wave to correlate the shear modulus. The described method has advantages on the hard material and multi-layered material. The manuscript should address the following issues before further consideration:

1. This manuscript mentions that the vibration range of PZT is much smaller than the central wavelength of light. Authors should explain how to capture the elastic wave with the OCT system.
2. For phantoms and human tissues, how to avoid the influence of depth information on vibration frequency information? Given that those tissues absorb the ultrasound and axial displacement cannot be negligible. Do the authors think that the elastic wave is still the Rayleigh surface wave?
3. The coefficient of attenuation is provided in the phantom study, but it is not given in the human study. Please explain.
4. Do the authors think that equation (4) can be applied to both phantom and human study?
5. The test results in Figure 5 are very close to the finite element simulation, can you provide the specific parameters of the finite element simulation?
6. In the OCT image in Figure 5, there is almost no signal at the bone position, how to get the phase signal?
7. In Figure 6, the elastic velocity increases with the vibration frequency, while in Figure 5 the Rayleigh velocity decreases with the vibration frequency. Please explain.
8. In Figure 6, half wavelength of the Rayleigh surface wave seems to be about 0.5 mm in 20kHz, what is it in 500kHz?
9. The Lamb wave model was used in this study, what sampling frequency was used? I am assuming this sampling frequency may not be the OCT A-line rate. Please provide more details about Lamb wave model to avoid potential confusion.
10. This study shows the capability of the proposed system to detect multi-layer structures, it is also necessary to demonstrate the capability to detect side-by-side structures, for example, a structure with shear modulus variations on the x-y plane, not along z direction.
11. Authors are encouraged to provide more details in the supplementary materials, it would be helpful to the readers.

POINT BY POINT RESPONSE TO REVIEWER COMMENTS

Reviewer #1 (Remarks to the Author):

In this work, the authors present an optical coherence elastography technique to extend the frequency range of dynamic elastography from the kHz range to the MHz range. Typically, the frequency range of the elastic wave is considered to be limited by the A-line rate. The technique presented here takes advantage of this property by intentionally causing an aliasing artifact, then performing anti-aliasing demodulation to perform high-frequency excitations effectively. Using this technique, the authors then demonstrate a variety of applications by performing mechanical estimation in hard materials, soft materials, and multilayered tissue, such as cartilage on bone and in vivo human skin.

The study is very interesting, and the technique enables OCE to be used in a broad set of cases. However, there are certain areas where the article could be improved.

1. "...Now that the modulation component is separated from the carrier, the vibration amplitude... manifest itself in the amplitude of the Fourier component..."

This claim is fundamental to the article, as it determines how vibration amplitude is detected in this work, and the method is different from how vibration amplitude is typically detected in phase-sensitive OCE. Later, the authors build on the claim with the following:

"...The side lobes were lower by 20 dB in signal power (10 dB in amplitude) than the main lobe. Since the power ratio is $20 \log_{10}(k_0 \delta)$, we get $\delta = 20.4 \text{ nm}$..."

These claims should be validated, showing that the measured vibration amplitude corresponds to the actual. Validation could be easily performed on a mirror mounted on a PZT.

Response 1: Thanks for the comment. We performed an additional experiment using laser Doppler vibrometer to validate our method of vibration amplitudes. Our results (see Fig. R1) show that the vibration amplitudes measured by our OCE method are in excellent agreement with those measured by Doppler vibrometer.

Changes made: We have described the details of the experiments and a figure (Fig. R1) in the Supplementary Materials.

Fig. R1 Validation of vibration amplitude measured by OCE with respect to a standard laser Doppler vibrometer. **a**, Experimental setup. Inset: zoom-in picture of the PZT. An adhesive retro-reflective tape was attached on the surface to enhance the vibrometer signal. **b**, The rf spectrum of a drive signal with an amplitude of 1.4 V and a frequency of 80 kHz. **c**, Measured vibrometer scan output for the given drive voltage in **b**. **d**, Linear slope efficiency of the PZT at 80 kHz. **e**, Comparison of the slope efficiencies measured by the laser vibrometer (red) and the OCE instrument (blue). The two datasets coincide within measurement accuracy.

2. The authors use a histogram to show the noise in the system, then describe a noise compensation technique to reduce the jitter. The authors should show a second histogram representing the difference before and after correction.

Response 2: To clarify, we did not reduce the time jitter of the laser, which originates from factors that could not be easily improved. Instead, our noise compensation technique corrects the *phase noise* induced by the time jitter. Therefore, the time jitter histogram remains unchanged.

Changes made: To avoid potential confusion, we have added a flowchart to the Supplementary Materials to clarify our noise compensation technique, as shown in Fig. R2. This flowchart demonstrates how we utilize the reference waveform to correct the phase noises induced by the laser time jitter. We also added the description of the noise compensation technique under “Noises and time jitter correction for ultrasonic frequencies” in Results.

Fig. R2 Correction of laser time jitter-induced phase noises. Channel 1 is the modulation waveform measured on the sample. Channel 2 is the reference waveform generated by the function generator and applied to the piezoelectric transducer (PZT). Both waveforms are recorded by the same data acquisition board. Channel 1 and Channel 2 are synchronized by the same trigger with a particular laser wavelength; thus they contain the same time jitters. The phase jitter over a sweep period (i.e. a M-scan containing m A-lines) is obtained by comparing the phase of the reference waveform with the theoretical phase. f_A is the A-line rate, f_m is the vibration frequency.

3. In Figures 2c and 2d, the x-axis labels seem to be flipped between the two images.

Response 3: In Fig. 2c, the x-axis represents the number of A-lines in each M-scan trace (in total 108). In Figure 2d, the x-axis represents the number of M-scan traces (in total 900). Thus, these two figures are not directly comparable.

Changes made: To make it clearer, we have revised the x-axis labels and the caption of Fig. 2 (now Fig. 3).

4. "...We obtained the phase of the modulation waveform (applied to the PZT) and subtracted it from the phase of the demodulated signal of each M-scan (Supplementary Note 1)..." This was not very clear from Supplementary Note 1.

Response 4: Thanks for pointing this out. The reference to 'Supplementary Note 1' should not be in this sentence, as the Note 1 describes the theoretical SNR-limited phase noise.

Changes made: We have removed "(Supplementary Note 1)" in this description. Also, we have modified our description of the noise correction technique in Results under "Noises and time jitter correction for ultrasonic frequencies".

5. In the results, the authors define the following equation, $z_m = \omega_m / (2k_1)$, then later in the

methods the equation is defined as $z_m = \omega_m / (2\kappa_1)$, where κ was replaced with κ . The authors should go back and ensure that the correct variables are used appropriately.

Response 5: Thanks.

Changes made: We have corrected the typo.

6. Supplementary Figure 9 should be moved to the main manuscript, as it's the clearest explanation of the author's technique. However, the figure can be improved. It's not immediately clear that Fig 9a) represents three unique signals; it almost seems like the same signal shifted. Additionally, in Figure 9b), there is no explanation in the caption for the gray dotted line in the caption or the figure. Showing the signal background that was subtracted might also be helpful.

Response 6: Thanks for this suggestion.

Changes made: We have improved Supplementary Fig S9 and moved it to the main manuscript (which is now new Fig. 2). This figure is shown below as Fig. R3.

Fig. R3 Simulation of anti-aliasing demodulation. **a**, A-line profiles of three scatterers at different depths for a modulation frequency of 688.8 kHz ($=16 \cdot f_A - 2.4$). The Fourier components at $F(\omega_0 - \omega_m)$, corresponding to $z_0 - z_m$, consists of three peaks: the vibration signal (negative sideband) originated from the scatterer at ω_0 or z_0 in depth (scatterer 1); the time-independent reflection signal (carrier signal) from a scatterer at depth $z_0 - z_m$ (scatterer 2); and the vibration signal (positive sideband) originated from the scatterer located at $z_0 - 2z_m$ (scatterer 3). The combined signal is shown. **b**, Steps of signal demodulation. (i) Original signal $F(\omega_0 - \omega_m)$. (ii) Static DC background signal $\langle F(\omega_0 - \omega_m) \rangle$. (iii) Subtracting the original signal from the background. (iv) Normalizing the signal by the background. (v) Fourier transform to identify $\delta(z_0)$ and $\delta(z_0 - 2z_m)$. Both the real part (red line) and the imaginary part (blue line) of the amplitude plots are shown. f_A : A-line rate, ω_0 : center wavenumber, ω_m : angular frequency of the vibration signal, δ : vibration amplitude.

7. Relating to the previous point, the organization of the paper could be improved. The first section of the results essentially summarizes the demodulation technique from a theoretical standpoint, which is further expanded in the methods, and the experimental procedure to actually perform the demodulation is relegated to the supplementary materials. Focusing more on the experimental procedure could make the work more impactful and clearer to a wider audience.

Response 7: We accepted this suggestion.

Changes made: We have added a new section “General demodulation algorithm for high frequency OCE” under Results to describe the experimental procedure to perform the demodulation as described in Fig. 2.

8. There are a few frequencies at play in this work, the modulation frequency, the A-line rate, and the carrier frequency (reflector). These are very vaguely defined and designated; correcting this could make the work easier to understand.

Response 8: Thanks for this helpful comment.

Changes made: We revised the manuscript to define these variables when they were mentioned for the first time in the manuscript.

9. Why was only the hydrogel data (Figure 4i) curve fitted?

Response 9: Sure, we added fitting results for the rubber and PDMS.

Changes made: We have included the fitting for the rubber and PDMS, as shown in Fig. R4.

Fig. R4 Fitting results for rubber and PDMS. For the rubber, a bi-linear fit is used for 0.2-20 kHz and 20-50 kHz separately due to the rapid increase of attenuation beyond 20 kHz.

10. The discussion section is quite bare. What are the limitations of the study? Some topics that could be discussed include the following:

High-speed OCT (such as FDML swept source OCT, ~3MHz sweep rate) could theoretically be used to obtain similar results using only the conventional OCE technique. The authors should discuss how the results may compare and the strengths and weaknesses of both approaches.

Moreover, high-speed OCT could be used to validate the proposed method at higher frequencies, something that was not done (but could/should be) in this research.

Response 10: Thank you for this suggestion. The integration of the ultra-fast OCT system into an ultrasonic OCE setup without aliasing is indeed a viable option. In both aliased and non-aliased schemes, all acquired data points are utilized for signal demodulation. Consequently, we anticipate comparable signal-to-noise ratio (SNR) for the same data acquisition time, with shot noise limiting the performance as long as practical sources of noise, such as laser phase noise, are minimized.

However, it is worth noting that ultrahigh-speed OCT systems with MHz A-line rates are currently available in only a few laboratories worldwide, and unfortunately, we do not possess one ourselves. Additionally, converting an OCT system into an OCE-capable system will require significant modifications and troubleshooting. While demonstrating OCE using MHz A-line rates would undoubtedly be a valuable endeavor in the future, we do not deem it necessary for this particular manuscript.

In our revised manuscript, we have emphasized that our algorithm is a general method applicable to both aliased and non-aliased schemes, irrespective of wave frequencies and A-line rates. We have also provided explicit clarification regarding the accuracy of our algorithm, which we validated using laser Doppler vibrometry, and its effectiveness across a wide range of wave frequencies spanning below and above the A-line rate of our system. Both schemes, in principle, can achieve identical SNR and attain shot-noise limited performance, provided practical noise sources such as laser phase noise are minimized.

The aliased scheme can be seen as an enabling technique that eliminates the constraint on A-line rates, rather than a limitation.

Changes made: We added a paragraph in the first section of Results, which reads “It is possible to extend OCE to ultrasonic frequencies simply by employing ultrafast OCT systems with MHz A-line rates³⁵⁻³⁷ while satisfying $f_m < 0.5f_A$. Then, the same data processing method as described above is applicable. However, since most OCT systems, including commercial products, use A-line rates less than 100 kHz, it should be worthwhile to develop a general method for OCE that works even when $f_m > f_A/2$. Below we describe such a method, which takes advantage of signal aliasing.”

We also added a paragraph in Discussion, which reads: The demodulation scheme was highly effective and allowed the upper frequency limit to be no longer limited by the A-line rate. This relaxes the system requirement. The demodulation scheme should be applicable to any swept source OCT systems regardless of their A-line rates. Ultra-high speed OCT systems with ultrahigh A-line rate up to several MHz are available using laser techniques such as Fourier domain mode locking^{35,36} and stretched-pulse active mode locking³⁷, and they will be well suited to build into MHz-range OCE systems without aliasing. In both aliased and non-aliased schemes, all the acquired data points are used for signal processing. Therefore, in principle the same SNR is expected for the same data acquisition time, and we expect shot noise-limited performance in both schemes as long as practical noises, such as the phase noise of laser sweep, are minimized. Ultimately, the maximum frequency for the wave based OCE technique is limited by viscous damping in samples, which can make waves overdamped. The critical frequency seems to be at a few hundreds of kHz for soft materials.”

11. The authors describe in the introduction the problem with wave damping over short distances for high frequencies. Their own results show the same problem (Figure 5 and Figure 6), as the proposed technique can't effectively solve that problem. How might this impact the utility of the proposed technique?

Response 11: We agree that wave damping will ultimately limit the maximum frequency of OCE measurement.

Changes made: We have added the following text in Discussion: "Ultimately, the maximum frequency for the wave based OCE technique is limited by viscous damping in samples, which can make waves overdamped. The critical frequency seems to be a few hundreds of kHz in soft materials".

12. The authors compare their results to the available literature, but the comparison is unavailable for the PDMS, hydrogel, and cartilage.

Response 12: We found literature data for PDMS, hydrogel, and cartilage and added the information to the manuscript.

Changes made: We added the literature data in Results under "Dynamic shear analysis of uniform soft materials" and "Depth-resolved stiffness mapping of cartilage".

Reviewer #2

Major revision

This manuscript demonstrates a novel method to detect high-frequency elastic waves and reconstruct the mechanical properties accordingly. Conventional OCT utilizes the Fourier transform of the spectrum to acquire depth information, in this work, the authors used the frequency of the elastic wave to correlate the shear modulus. The described method has advantages on the hard material and multilayered material. The manuscript should address the following issues before further consideration:

1. This manuscript mentions that the vibration range of PZT is much smaller than the central wavelength of light. Authors should explain how to capture the elastic wave with the OCT system.

Response 1: OCE uses phase-sensitive interferometric detection to measure vibration amplitudes smaller than the central wavelength of light. The phase sensitivity is proportional to the signal to noise ratio (SNR) and the number of A-lines that are used for signal processing. With an SNR of 20 dB and 100 A-lines, OCE can measure a displacement of 1% of the optical wavelength. This principle is described extensively in literature.

Changes made: We have included additional descriptions of phase-sensitive detection in Results under "Experimental demonstration of anti-aliasing demodulation": "We used phase-sensitive interferometric detection^{38,39} to obtain sub-wavelength vibration amplitudes."

2. For phantoms and human tissues, how to avoid the influence of depth information on vibration frequency information? Given that those tissues absorb the ultrasound and axial displacement cannot be negligible. Do the authors think that the elastic wave is still the Rayleigh surface wave?

Response 2: We apologize but we do not understand the question so clearly. As we demonstrated, we can actually take advantage of the relationship to extract depth-dependent characteristics from the measured vibration frequency information.

The elastic waves on or near the surface would always have axial components or displacement parallel to the propagation direction, even without acoustic attenuation. However, this does not affect the validity of our principle.

The Rayleigh surface wave does have axial displacement, but the OCE technique measures the transverse displacement (orthogonal to the surface) to determine the wave wavelength.

3. The coefficient of attenuation is provided in the phantom study, but it is not given in the human study. Please explain.

Response 3: We used an elastic model to approximate the skin layers rather than the viscoelastic model used for the bulk isotropic phantom. This is because the elastic modulus of epidermis is greater than dermis and hypodermis, so the surface waves are leaky waves. This anti-waveguiding loss from the leaky waves makes it difficult to extract the intrinsic material absorption or the coefficient of attenuation from measured wave profiles.

Changes made: We have explained why we used an elastic model for the skin layers in Results under “Depth-resolved stiffness mapping of *in vivo* human skin”.

4. Do the authors think that equation (4) can be applied to both phantom and human study?

Response 4: Yes, equation (4) can be applied to both phantom and human study. In a bulk isotropic material, the bulk shear velocity is independent on frequency, thus $v' = dv/df = 0$. Inserting $v' = 0$ into equation (4) results in $\mu = \rho v^2$.

Changes made: We have added the following text in Results under “Depth-resolved stiffness mapping of *in vivo* human skin”: “Notice that in bulk isotropic materials, where $v' = 0$, Eq. (4) is reduced to $\mu = \rho v^2$.”

5. The test results in Figure 5 are very close to the finite element simulation. Can you provide the specific parameters of the finite element simulation?

Response 5: Thanks for the suggestion.

Changes made: We have added the specific values used for the parameters in Methods under “Finite element analysis”.

6. In the OCT image in Figure 5, there is almost no signal at the bone position, how to get the phase signal?

Response 6: The bone is opaque, so the optical signal is very low at the bone position, so we cannot get the phase signal from within the bone. Nonetheless, Substantial energy of elastic waves reach the bone, partially reflected from and partially transmitting into the bone. This boundary effect of elastic waves affects the wavelength of the wave along the surface and is reflected on the wave dispersion curve.

Changes made: To clarify this point, we have added a dashed line to label the interface between the calcified cartilage (CC) and the bone in Figure 6d and 6e. We have added the description of bone signals under “Depth-resolved stiffness mapping of cartilage” in Results.

7. In Figure 6, the elastic velocity increases with the vibration frequency, while in Figure 5 the Rayleigh velocity decreases with the vibration frequency. Please explain.

Response 7: It is because of the opposite depth-dependent profiles of the two different samples. In some way, we have deliberately chosen those samples to illustrate the striking relationship between the wave dispersion and the sample characteristics.

Changes made: To make this important point clear, we have added a Supplementary figure, as shown in Fig. R5, to explain the differences in the wave dispersion relations. We also added relevant descriptions in Results under “Depth-resolved stiffness mapping of *in vivo* human skin”.

Fig. R5 Penetration of Rayleigh surface waves. **a**, Isotropic material. **b**, Multilayer material with softer top and stiffer bottom (e.g., the cartilage). **c**, Multilayer material with stiffer top and softer bottom (e.g., the skin). E_1 , E_2 , and E_3 represent the elastic moduli of the three layers. Lower graphs represent dispersion curves. In isotropic materials, the phase velocity is independent of the stimulus frequencies (f_1 , f_2 , and f_3). For multilayer materials, the phase velocity varies with frequency as the penetration depth of the Rayleigh wave varies with frequency. The amplitude of Rayleigh wave is reduced to $1/e$ is at a depth of about a half wavelength.

8. In Figure 6, half wavelength of the Rayleigh surface wave seems to be about 0.5 mm in 20khz, what is it in 500khz?

Response 8: The half wavelength of the Rayleigh surface wave is about 0.2 mm at 500 kHz, and about 1.1 mm at 20 kHz.

Changes made: In the caption of Fig. 7b, we added the half wavelength of the Rayleigh surface wave for 500 kHz and 20 kHz.

9. The Lamb wave model was used in this study, what sampling frequency was used? I am assuming this sampling frequency may not be the OCT A-line rate. Please provide more details about Lamb wave model to avoid potential confusion.

Response 9: We apologize, but it is not clear what sampling frequency is referred to here. The data sampling frequency from the photodetectors is 100 MHz, as stated in Methods. This sampling frequency is not related to Lamb wave models.

Changes made: We have included a few sentences and additional references about Lamb waves.

10. This study shows the capability of the proposed system to detect multi-layer structures, it is also necessary to demonstrate the capability to detect side-by-side structures, for example, a structure with shear modulus variations on the x-y plane, not along z direction.

Response 10: It is possible to use the technique to detect variations in the x-y plane. In a recent study, we used the same OCE system to investigate the transverse variation in the elastic properties of the cornea in human subjects. Figure R6 below shows the result, which reveals the remarkable gradient of the elastic wave velocity from the sclera to the central cornea.

Fig. R6 Mapping the stiffness of human eye using OCE. **a**, Monitoring camera view of the eye. The OCT beam scan path is marked by a dotted line. **b**, 2-D phase velocity map reconstructed with a spatial resolution of ~ 0.5 mm.

Changes made: We described the capability of our technique for mapping stiffness variations in the x-y plane in the last paragraph of Discussion.

11. Authors are encouraged to provide more details in the supplementary materials, it would be helpful to the readers.

Response 11: Thanks for the suggestion.

Changes made: We have added 3 Supplementary figures (S3, S4, and S10) to provide more details about our study.

Reviewer #1 (Remarks to the Author):

The authors have addressed many of the comments that were brought up. The addition of Figure 2 and Supplementary Figure 4 are particularly appreciated. Minor comments as follows:

1. Fig S4 greatly clarifies the method being discussed. Can the authors expand on how the reference waveform fed to the PZT is measured through OCT? Presumably, the authors are feeding the signal to the system without causing the PZT vibration; if so, what exactly is being measured?
2. Could the authors indicate that the last plot in Figure 2a is the additive figure, consisting of components of the three scatters? This way it would be clearer that the first three plots are describing something that is seen in the last.
3. The authors describe an alternative method to subtracting the DC background by adding a large constant bias. An addition to Figure 2 to clearly show how that process would work and feed into the next step would greatly clarify the method.

Reviewer #2 (Remarks to the Author):

The author has revised all concerns reviewer asked. It was suggested to accept it.

POINT BY POINT RESPONSE TO REVIEWER COMMENTS

Reviewer #1 (Remarks to the Author):

The authors have addressed many of the comments that were brought up. The addition of Figure 2 and Supplementary Figure 4 are particularly appreciated. Minor comments as follows:

1. Fig S4 greatly clarifies the method being discussed. Can the authors expand on how the reference waveform fed to the PZT is measured through OCT? Presumably, the authors are feeding the signal to the system without causing the PZT vibration; if so, what exactly is being measured?

Response 1: To clarify, the reference waveform refers to the stimulus waveform fed to the PZT (shown as the green curve in Fig. S2) and was recorded through the analog input channel of the input/output board rather than measured through OCT. The reference waveform and the modulation waveform are both recorded simultaneously during the vibration of the PZT to guarantee that they contain the same time jitters.

Changes made: To avoid potential confusion, we have replaced “reference waveform” by “stimulus waveform” throughout the manuscript. Additionally, to clarify we have modified the description of the time jitter correction in the main manuscript and supplementary Figure S4.

2. Could the authors indicate that the last plot in Figure 2a is the additive figure, consisting of components of the three scatters? This way it would be clearer that the first three plots are describing something that is seen in the last.

Response 2: Thanks for this suggestion.

Changes made: We have added “Combined” in the additive figure in Figure 2a and also added a statement in the caption that the last plot represents the combined signal of all three scatterers. Additionally, we made minor modifications in Figure 2b to better clarify the DC signal.

3. The authors describe an alternative method to subtracting the DC background by adding a large constant bias. An addition to Figure 2 to clearly show how that process would work and feed into the next step would greatly clarify the method.

Response 3: In the alternative method, we also need to obtain the DC background to move the phasor to the real axis prior to adding a large bias. Therefore, the alternative method is essentially equivalent to the original method.

Changes made: To simplify our algorithm, we deleted the description of the alternative method in the main manuscript.

All concerns are addressed

POINT BY POINT RESPONSE TO REVIEWER COMMENTS

REVIEWERS' COMMENTS

All concerns are addressed

Response: We thank all the reviewers for their helpful suggestions and comments.